. Pathogens

# High-resolution genome assembly and linkage mapping in *Meloidogyne hapla* reveal non-canonical telomere repeats and recombination hotspots associated with effector proteins

Pallavi Shakya[1], Muhammad I. Maulana[2,3], Etienne G. J. Danchin[4], M. Laurens Voogt[2], Stefan J. S. van de Ruitenbeek[2], Jacinta Gimeno[1], Adam P. Taranto[1,5], Alison C. Blundell[1], Evelin Despot-Slade[6], Nevenka Meštrović[6], Ana Zotta Mota[4], Dadong Dai[7], Valerie M. Williamson[1], Mark G. Sterken[2]*, Shahid Siddique[7]*

**1** Department of Plant Pathology, University of California, Davis, California, United States of America, **2** Laboratory of Nematology, Droevendaalsesteeg PB, Wageningen University and Research, Wageningen, The Netherlands, **3** Department of Plant Protection, Faculty of Agriculture, Universitas Gadjah Mada, Yogyakarta, Indonesia, **4** INRAE, Université Côte d'Azur, CNRS, Sophia-Antipolis, France, **5** School of BioSciences, The University of Melbourne, Melbourne, Victoria, Australia, **6** Ruđer Bošković Institute, Zagreb, Croatia, **7** Department of Entomology and Nematology, University of California, Davis, California, United States of America

* mark.sterken@wur.nl (MGS); ssiddique@ucdavis.edu (SS)

## Abstract

Root-knot nematodes (*Meloidogyne spp.*) are among the most destructive agricultural pests that cause significant yield losses across a wide range of crops. *Meloidogyne hapla* is a valuable model for studying root-knot nematodes due to its parasitic diversity, small diploid genome, and a reproductive strategy that facilitates genetic analysis. Here, we report the most contiguous genome assembly to date for any plant-parasitic nematode built using PacBio HiFi, Oxford Nanopore, Illumina, and Hi-C sequencing. Genetic linkage analysis of F2 populations derived from crosses between *M. hapla* strains validated the assembly but also revealed anomalies indicating chromosome structure differences between parental isolates such as fissions, fusions, and rearrangements. Strikingly, we identified sharply delimited zones with extraordinarily high recombination on most chromosomes. Notably, several of these high recombination zones were significantly enriched for genes encoding secreted proteins, many of which contribute to parasitism. These findings suggest that meiotic recombination facilitates effector diversification and offer insight into how these parasites diversify their effector protein repertoire to change or expand their extraordinary host range. We further report the discovery of a novel 16-nucleotide tandem repeat and lack of canonical telomere repeats at chromosome ends. The localization of this 16-nt repeat at chromosome ends highlights a potentially divergent mechanism of chromosome-end maintenance in this nematode group. Overall, our study integrates high-resolution structural genomics, genetic mapping, and functional inference to

the Creative Commons Attribution License, which permits unrestricted use, distribution, and reproduction in any medium, provided the original author and source are credited.

**Data availability statement:** All sequencing data have been deposited in the Sequence Read Archive (SRA) at the National Center for Biotechnology Information (NCBI). The HiFi, Nanopore, Illumina, HiC, and Iso-Seq data were deposited under Bioproject PRJNA1265270. The processed genome and annotation data is available under Bioproject PRJNA1265270. The supplementary and source data is public-ly available at Zenodo research repository (https://doi.org/10.5281/zenodo.15484846). The scripts used for generation of figures and data analysis can be found in: https://github.com/Siddique-Lab/Mhapla_genome_paper; https://git.wur.nl/stefan.vanderuitenbeek/rnaseq_variant_calling_snakemake_pipeline/;https://git.wur.nl/published_papers/vw9_genome.

**Funding:** The research performed in the laboratory of S.S. was supported by a grant from the National Science Foundation (NSF; IOS 2203286). The work conducted in the laboratory of M.G.S. was supported by a grant from the Dutch Research Council (NWO; VENI grant 17282). The funders had no role in study design, data collection and analysis, decision to publish, or preparation of the manuscript.

**Competing interests:** The authors have declared that no competing interests exist.

uncover links between genome architecture, recombination landscapes, and host–parasite interactions.

---

## Author summary

Root-knot Nematodes (RKNs) are major agricultural pests, causing widespread crop losses globally. Among them, *Meloidogyne hapla* is a particularly valuable research model due to its compact diploid genome and reproductive flexibility that enables genetic studies. In this study, we generated a complete and ac-curate chromosome-scale genome assembly of *M. hapla* using state-of-the-art sequencing technologies. We used genetic maps to validate the accuracy of the genome and found that some *M. hapla* strains have structural differences in their genome wherein the chromosomes have fused or broken apart. We also discovered that *M. hapla* has zones with extraordinarily high recombination rates on most of its chromosomes and these zones are enriched in predicted secreted peptides that may contribute to parasitism. This suggests that recombination may help these nematodes to evolve new ways of overcoming plant defenses. Finally, we identified an unexpected 16-nucleotide long repeat at chromosomal ends instead of a typical telomere sequence hinting to an alternative strategy of chromosomal maintenance. Overall, our study provides the fundamental genom-ic resources on *M. hapla* and reveals how the genome structure and recombina-tion have shaped the parasitism in this organism.

## Background

Nematodes are among the most abundant and diverse animal phyla, containing an estimated 1–10 million species [1]. This remarkable diversity is reflected in their eco-logical niches—ranging from free-living forms in soil and water to parasitic species infecting plants and animals. Among the approximately 4,100 species identified as plant parasites, species in the genus *Meloidogyne* (commonly known as root-knot nematodes, or RKNs) stand out as the most economically damaging [2,3]. The four most damaging species, *M. arenaria*, *M. incognita*, *M. javanica*, and *M. hapla,* pos-sess an extraordinary host range spanning a diverse array of crop species [4]. Three of these species (*M. arenaria*, *M. incognita*, *M. javanica*) are closely related (*Meloid-ogyne* Clade I) and are globally distributed in tropical and subtropical regions. In con-trast, *Meloidogyne hapla* (Clade II) is generally found in more temperate climates and parasitizes a broad range of host plants, although isolates differ in their host range, pathogenicity, and behavior [5–8].

The RKN lifecycle generally spans about a month and encompasses six distinct stages consisting of embryo, four juvenile stages (J1-J4) and an adult stage [9,10]. As obligate sedentary endoparasites, RKNs spend most of their life cycle feeding from a permanent site within the root vascular system. These feeding sites are

characterized by nematode-induced multinucleated giant cells that act as nutrient sinks for the nematodes, and by distinctive root galls or "knots", a hallmark of RKN infestation [11,12]. How RKNs establish these specialized feeding sites in such a broad range of plant species, spanning monocots, dicots, annuals, and perennials is a question of both scientific and practical interest. Nematode genes responsible for establishing feeding sites or contributing to differences in host range are largely unknown. However, several proteins secreted by plant-parasitic nematodes have been demonstrated to contribute to parasitism and are commonly referred to as effectors [13–16]. These effectors include genes likely acquired via horizontal transfers as well as many encoding pioneer proteins with no known protein motifs.

*Meloidogyne* spp. exhibit a wide range of karyotypes and reproductive mechanisms [17]. Whereas most Clade I species reproduce asexually without meiosis and carry genomes with various degrees of polyploidy, most isolates of *M. hapla* are diploid and reproduce by facultative meiotic parthenogenesis [5,18]. Sexual reproduction occurs when females are fertilized by migratory males, and sperm-oocyte fusion occurs to generate offspring [18,19]. However, cytological studies have shown that, in the absence of males, sister chromatids of meiosis II are rejoined to restore diploidy asexually [18,19]. This reproductive mechanism has facilitated controlled crosses between strains and the production of F2 lines. Molecular marker analysis of F2 lines from a cross of two different strains showed these lines are largely homozygous for sequence polymorphisms and thus resemble recombinant inbred lines (RILs) [19]. Additionally, these F2 lines have been used to produce a marker-based genetic map and to identify genetic loci affecting interactions with host and/or behavior [20–23].

Our study focuses on *M. hapla* for its genetic tractability and diploid genome, which make it a favorable model and therefore reference organism for RKN genetics and genomics. A previous draft genome sequence of the inbred *M. hapla* strain VW9 has been produced, and a DNA-based linkage map was generated based on segregation of polymorphisms in RIL-like F2 lines from a cross between strains VW9 and VW8 [24]. However, this genome assembly is highly fragmented, limiting both the localization of genes responsible for phenotypic traits and synteny analyses with other RKNs. Recent advances in sequencing technologies have led to highly contiguous assemblies for several diploid and polyploid *Meloidogyne* species [25–28]. In these studies, the canonical nematode telomeric repeats (TTAGGC)$_n$ were not detected at chromosome ends or anywhere else in the assemblies. Instead, in three Clade I species, *M. incognita*, *M. arenaria* and *M. javanica,* long arrays of species-specific complex tandem repeats were found, mostly enriched at one scaffold end [26,27]. Furthermore, telomere associated proteins such as telomerase and shelterin complexes that are evolutionarily conserved in other nematode clades were not identified suggesting an alternative mechanism for chromosome end maintenance may be at play in RKN [27].

Here, we generated a *de novo* chromosome-level assembly of diploid *M. hapla* using complementary long-read sequencing strategies. We characterized another non-canonical repeat sequences at the end of chromosome-length scaffolds. Additionally, we utilized data from previously generated F2 lines to compare scaffold structure with genetic linkage groups. This analysis revealed a recombination landscape characterized by extraordinary recombination rates mostly on chromosome arms and provided evidence for differences in chromosome structure/behavior between isolates. To the best of our knowledge, this is the first chromosome-scale RKN genome assembly validated by genetic linkage analysis. Furthermore, we examine the chromosomal distribution of genes encoding secreted proteins (effector candidates) and provide evidence for their enrichment in the high recombination zones.

## Results

### Chromosome-level genome assembly of *Meloidogyne hapla* strain VW9

We produced a high-quality, chromosome-level genome assembly for *Meloidogyne hapla* strain VW9, the same strain used in a previous assembly by Opperman et al. [24] (**PRJNA29083**). We used a combination of PacBio HiFi sequencing (56x coverage), Oxford Nanopore Technologies (ONT) long-read sequencing (143x coverage), Illumina short read sequencing (138x coverage), and Hi-C Chromatin conformation capture (263x coverage). Using Hifiasm with the HiFi,

ONT, and Hi-C datasets, we produced an initial assembly with 36 contigs. This assembly was assessed for contaminants with Blobtools, where all 36 contigs were assigned the taxonomy Nematoda (S1 Fig). This assembly was further polished with the Illumina reads and subsequently scaffolded using Juicer and 3D-DNA with the Hi-C data (Fig 1A). Additionally, the mitochondrial genome was assembled (S2 Fig). The resulting assembly consisted of 16 scaffolds for a total length of 59.2 Mb with an N50 of 3.8 Mb. The assembly size closely matched the 61.6 Mb genome size estimated by k-mer analysis (S3A Fig) and was consistent with flow cytometry data, which estimated the diploid nuclear DNA content of *M. hapla* to be $121 \pm 3$ Mb, corresponding to a haploid genome size of approximately 60 Mb [29]. Analysis of the Illumina data supported a diploid genome structure for *M. hapla* and almost entirely homozygous (S3B Fig). The high integrity of this assembly was further supported by the Merqury k-mer plot, which indicated that almost all the information present in Hi-Fi reads was captured in the haploid assembly (S3C Fig). Overall, this new *M. hapla* genome assembly represents a significant improvement over the previous version and is the most contiguous so far for a RKN.

In the Hi-C contact map, we observed patterns suggesting physical proximity between scaffold tips of different chromosomes (**Fig 1A**). To assess whether these patterns were an assembly artifact, we checked the quality of raw Hi-C reads and built Hi-C contact maps with different draft assemblies (S4 Fig). This analysis supports that the patterns are due to inter-chromosomal tip proximity.

To define the scaffold ends in the *M. hapla* assembly we analyzed the terminal 24 kb of each scaffold for tandem repeats using Tandem Repeat Finder (TRF) [30]. This analysis revealed arrays of a 16-mer with the consensus sequence (CCCAAGGTTTAAAAGG) at 17 scaffold ends and a variant 16-mer detected near the end of Scaffold 10. This 16-mer repeat was the only significant repeat detected in the terminal regions by TRF. Nine scaffolds (S3, S4, S5, S6, S7, S12, S13, S15, S16) have this tandem repeat at only one end, while four scaffolds (S2, S8, S9, and S14) show the same tandem repeat at both ends (**Fig 1B**). The length profile of these arrays ranges from approximately 1,289 bp, to 8,912 bp. For S10, a variant tandem repeat with consensus TTATAAAGGAAGTGGG is present starting 4 kb from one end. A search for the 16-mer tandem repeat array within scaffolds identified only three arrays at the center of the S1 scaffold (Fig 1B and S1 Table). The internal arrays in S1 form a large palindrome with over 300 copies of the repeat followed by over 300 copies of the complementary sequence. Among the scaffolds, only S11 entirely lacked either repeat array.

In addition, to validate whether these candidate telomeric repeats indeed localize to the chromosome ends, we performed fluorescence in situ hybridization (FISH) analysis on elongated chromosomes of *M. hapla* (**Fig 2**) using the 16-mer repeat found at 17 scaffold ends. Due to the tiny size of the chromosomes and the difficult biological material, we could not obtain complete chromosome complement spreads. Nevertheless, for the intact chromosomes that were discernible, the 16-mer repeats predominantly localized at or near the termini of multiple chromosomes (**Fig 2A and 2B**). Several chromosomes showed signal at only one end, while others displayed no signal (**Fig 2B**). The signal intensity varied between chromosomes, indicating differences in repeat array lengths. Furthermore, we detected a signal internally for at least one chromosome (**Fig 2A**), corroborating the presence of an internal repeat array as observed in S1. Overall, FISH results confirm the 16-mer repeat is enriched predominantly at chromosome ends and most likely constitute specific telomeric sequences in *M. hapla*.

After identifying and validating the primary 16-mer repeat, we searched for additional repeat motifs at scaffold ends using MEME suite [31]. The analysis revealed two distinct motifs: one at the ends of chromosomes 3, 10, 13 and 15, and another at ends of chromosomes 1 and 11 (S5 Fig and S1 Table). These motifs remain to be independently confirmed through FISH validation. Additionally, we searched for these repeats in the raw ONT reads using TelosearchLR [32]. We identified the primary 16-mer repeat as the top-ranked motif. Other motifs were either variants of the 16-mer or non-terminal repeats not enriched at chromosome ends. These results validate the 16-mer as the dominant telomere-associated repeat in *M. hapla* VW9.

Within the nematode order Rhabditida, the availability of improved chromosome-length assemblies led to the identification of seven ancient linkage blocks, known as Nigon elements [33]. These elements corresponded to six

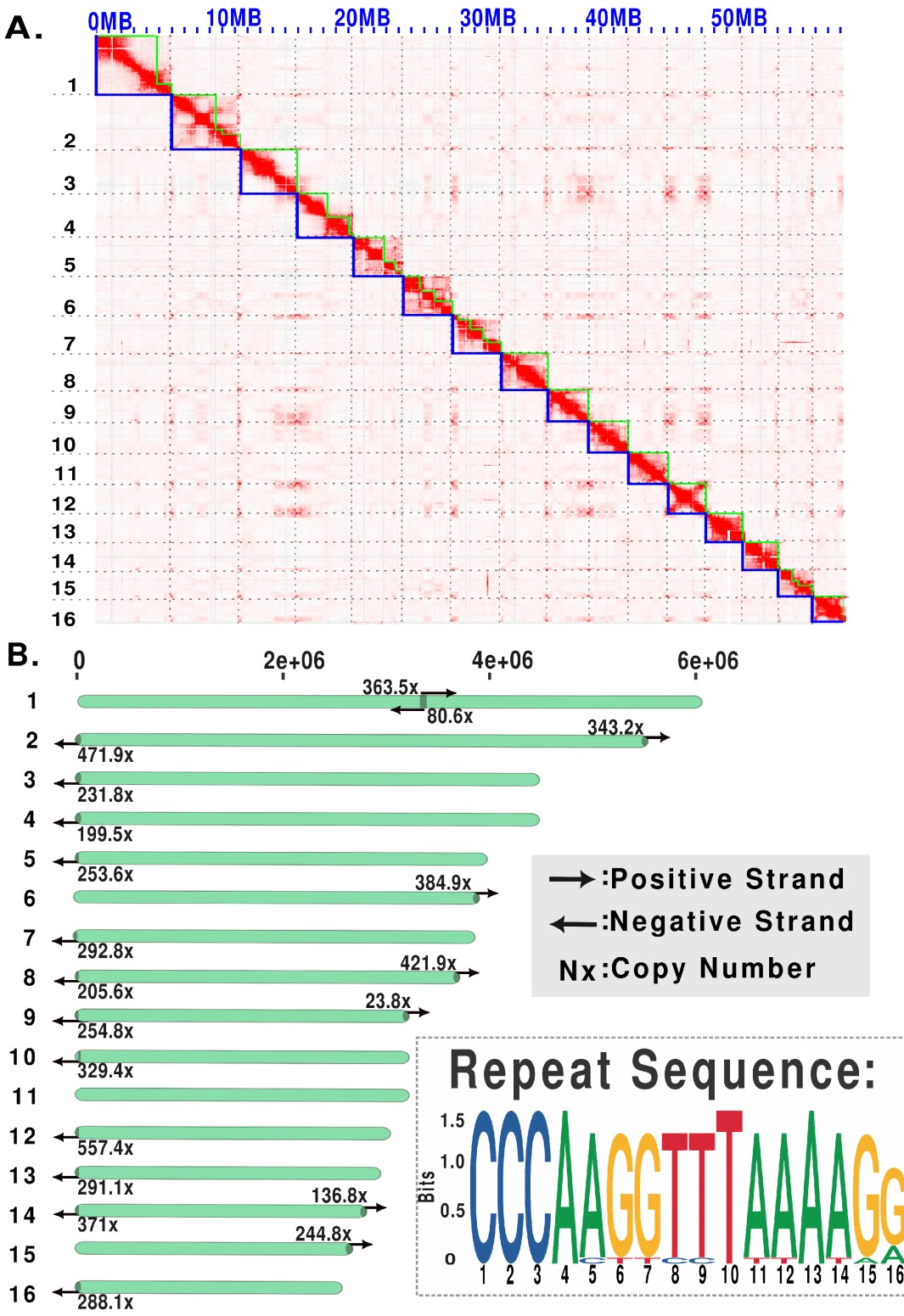

**Fig 1. Chromosome-scale genome assembly and repeat structure of *Meloidogyne hapla* strain VW9.** A. HiC contact map of *M. hapla* showing 16 chromosome-scale scaffolds. Green lines denote the edges of contigs, and blue lines denote the edges of scaffolds. B. Distribution of the 16-mer repeats across chromosome-scale scaffolds. Each horizontal bar represents a scaffold, with arrows indicating repeat orientation (rightward: positive strand; leftward: negative strand) and numbers showing repeat copy number per scaffold. The repeat at the end of Scaffold 10 is a variant of the others shown.

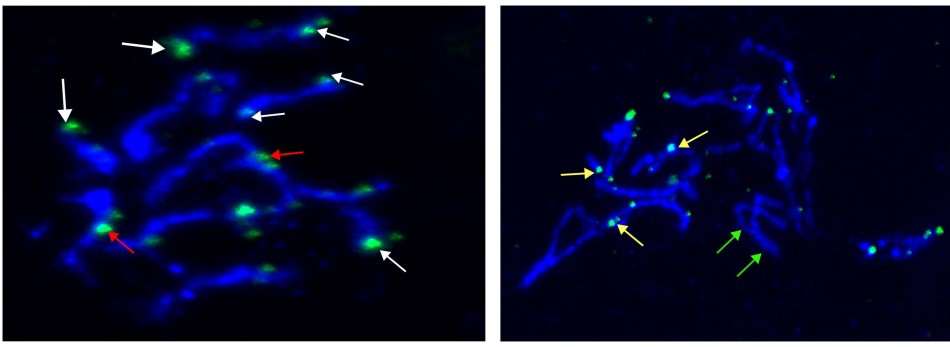

**Fig 2. DNA FISH with the 16 bp tandem repeat probe on *M. hapla* chromosomes in different chromosome condensation stages (A and B).** Probe is labeled with FITC (green) and chromosomes are counterstained with DAPI (blue). Arrows point to hybridization signals localized at one (white arrows) or both (yellow arrows) chromosome ends. Red arrow indicates a chromosome where tandem repeats appear to be located internally. Green arrows point to chromosomes without hybridization signals. Size bar = 5 μm.

autosomes (A, B, C, D, E, N) and a sex chromosome (Nigon X), based on the co-localization of orthologous gene groups. These Nigon elements have been identified in multiple nematode species, including *Caenorhabditis elegans*, *Pristionchus pacificus*, and *Auanema rhodensis* [34–36]. Using the previous *M. hapla* assembly, no clear Nigon assignments were identified, likely due to the highly fragmented nature of the assembly [33]. However, analysis of the current *M. hapla* assembly identified fragments of Nigon elements, with detectable, although highly fragmented, chromosomal segments. Nigon X, the ancestral sex chromosome, was not clearly identifiable (**Fig 3**). This pattern of Nigon elements suggests that extensive chromosomal rearrangements have occurred since *M. hapla* diverged from the ancestral 7-chromosome rhabditid species.

**Genetic linkage map supports physical assembly and implies structural variation between nematode isolates**

To assess whether our chromosome-length scaffolds correspond to true chromosomes, we produced a scaffold-based linkage map utilizing expressed sequence SNPs (eSNPs) from F2 lines derived from a cross between *M. hapla* strains VW9 and LM [20]. Using the new assembly, we identified 789 SNP alleles flanking crossover events in each of 84 F2 lines (**Fig 4**). The genome-wide recombination rate was high – 14.7 cM/Mb – compared to the 2.9 cM/Mb found in *C. elegans* [37]. The recombination frequency profiles (Marey maps) for twelve scaffolds (S2, S3, S5, S6, S8, S9, S10, S11, S13, S14, S15, S16) are characterized by a high recombination rate in the chromosome arms and low recombination in the chromosome centers or tips (**Fig 5**). This recombination pattern is similar to those described for *C. elegans*, *C. briggsae*, and *P. pacificus* [37,38]. However, the remaining chromosomes showed intriguing deviations from the typical recombination profile. Most notably, S1 showed high recombination through the central region in addition to chromosomal arms, while S4 showed no recombination and S7, S12 and S13 showed low recombination. Intriguingly, segregation correlation in F2 lines suggested that S1 consisted of two linkage groups and that S4 and S12 alleles were on the same linkage group (**S6 Fig**). In addition, S13 showed highly skewed distribution of alleles in F2 lines favoring LM (**Fig 4**).

We used change point analysis to further define the distribution and boundaries between regions with high and low recombination. This revealed sharp differentiation along the scaffolds between high recombination zones (HRZs), where rates ranged from 11 to 139 cM per Mb, and low recombination zones (LRZs) where rates were <5 cM/Mb (**Fig 5** and **S2 Table**).

To test whether scaffold mis assembly was contributing to the idiosyncrasies that we observed in the scaffold-based linkage map, we produced a *de novo* classical genetic linkage map using segregation data from 458 SNPs in 93 RIL-like F2 lines from the previously described cross between *M. hapla* strains VW9 and LM [20]. The resulting map contained 19

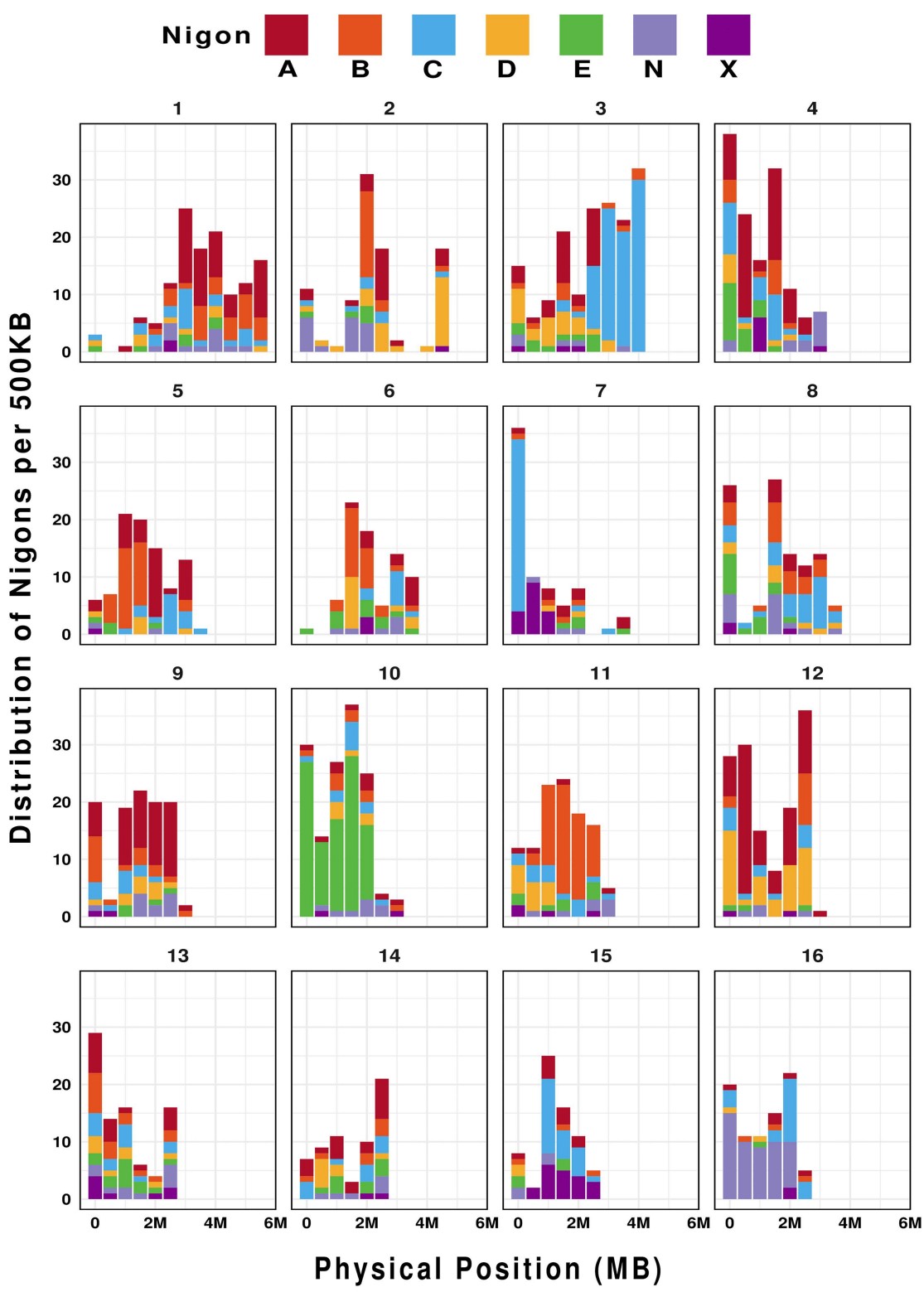

**Fig 3. Distribution of Nigon elements along the length of *M. hapla* scaffolds.** The X-axis represents the physical length of each scaffold, and the Y-axis represents the Nigon-defining loci per 500 Kb non-overlapping windows. The legend shows the color key for each Nigon Element from A through X.

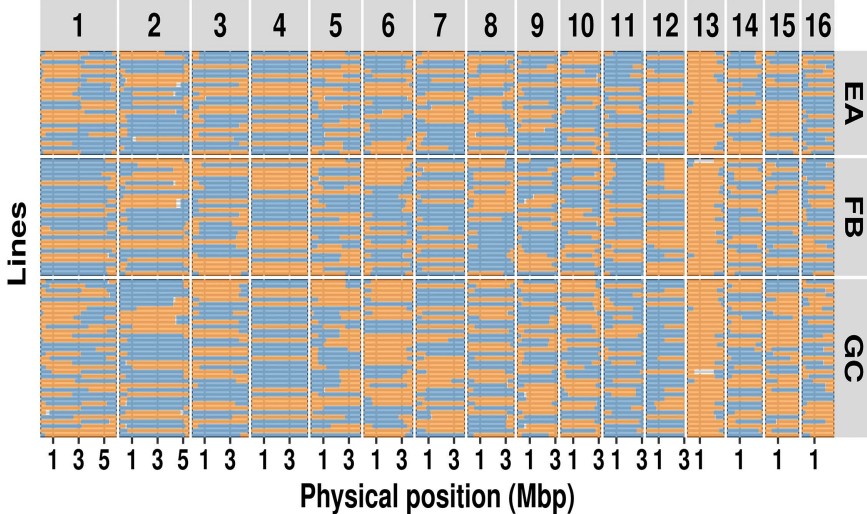

**Fig 4. Recombination profile based on chromosome scaffolds.** Allele present in each F₂ line on scaffolds 1-16. Colors represent regions with VW9 (blue) and LM (orange) alleles, whereas white areas indicate insufficient data. Panels EA, FB, and GC on the right indicate the female from which the F₂ lines were derived.

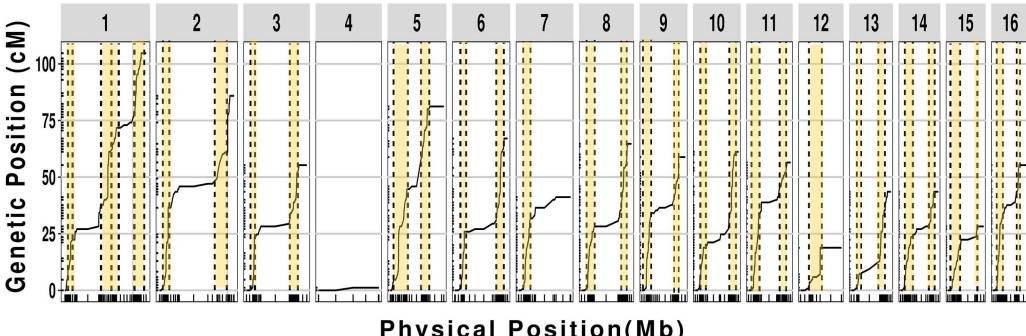

**Fig 5. Recombination profile for each scaffold.** Tick marks on the x axis indicate physical position on the scaffold and those on the y axis are the corresponding genetic positions based on SNP segregation in F₂ lines. High recombination zones (HRZs) are highlighted in yellow. The number of scaffolds is depicted on top of each box.

linkage groups (LGs). We then located the mapped SNPs on scaffolds in the new genome assembly to compare chromosome scaffolds to the genetic linkage groups (S3 Table). For thirteen linkage groups, all SNPs were assigned to a single scaffold, supporting that these scaffolds represented full length chromosomes (Table 1). However, scaffolds S1, S2 and S13 were each divided into two genetic linkage groups. The most striking discrepancy was S1, which formed two linkage groups in the classical map.

As noted above, S1 showed anomalously high recombination in the center as well as on arms. Interestingly, the break between the linkage groups in the VW9xLM cross and the high central recombination occurred in the region of the internal palindrome corresponding to the candidate telomeric repeat (**Fig 1B**). To determine whether this joining was a scaffolding error, we examined nanopore reads and identified 2902 raw ultralong nanopore reads greater than 60 kb spanning the palindromic region of S1. Hence, the existence of a single molecule spanning this region is strongly supported. Additionally,

**Table 1. Alignment of *Meloidogyne hapla* VW9 scaffolds with genetic linkage groups.**

| Scaffold | Genetic Linkage Group | |
|---|---|---|
| | VW9 X LM[a] | VW8 X VW9[b] |
| S1 | LG1a, LG1c | LG1 |
| S2 | LG2a, LG2c.15 | LG2, LG15 |
| S3 | LG4 | LG4 |
| S4 | LG14 | LG14, LG17 |
| S5 | LG10.12 | LG10, LG12 |
| S6 | LG2b | LG2 |
| S7 | LG6 | LG6 |
| S8 | LG3 | LG3 |
| S9 | LG1b | LG1 |
| S10 | LG16 | LG16 |
| S11 | LG5 | LG5 |
| S12 | LG13.17 | LG13 |
| S13 | LG9a, LG9b | LG9 |
| S14 | LG7 | LG7 |
| S15 | LG8 | LG8 |
| S16 | LG11 | LG11 |

[a]LG names are based on those assigned in previous work [20,22].

we noted a difference between the recombination patterns in progeny from female FB, one of the three F1 females from which the RIL-like F2 lines derive showed no recombinants in the central region of S1 chromosome whereas progeny from the other two females (EA and GC) showed high apparent segregation in this region.

To further investigate the anomalies between scaffolds and linkage groups, we utilized a published genetic linkage map generated with segregation data of DNA polymorphisms in 183 F2 lines from a cross between *M. hapla* strains VW8 and VW9 [21]. We then located the mapped SNPs onto scaffolds in our new genome assembly. Ten linkage groups corresponded to single scaffolds (**Table 1**). However, even though VW9 was a parent in both crosses (VW9xLM and VW8xVW9), there were differences in the alignment of chromosome scaffolds to linkage groups for the two crosses. Notably, LG1 spanned all SNPs of S1 and S9, predicting it to be a very long linkage group. Also, LG2 contained all markers for scaffolds S2 and S6. Interestingly, the highly skewed marker segregation in favor of LM alleles seen for S13 in the VW9xLM cross was not observed in the VW8xVW9 cross. In addition, in the VW8x9 linkage map, S4 and S12 did not show repressed recombination. Since both genetic crosses used VW9 as one parent but a different second parent, the disparity between the chromosome scaffolds and linkage maps is likely due to structural differences between genomes of the three strains. For example, inversions, translocations or other genomic rearrangements (chromosome fusions, breakage) could result in the distorted segregation patterns that we observed. Other intriguing discrepancies between scaffolds and LGs remain to be explained but would likely require producing chromosome-length sequences of multiple *M. hapla* strains.

Together these results suggest that the differences between linkage maps and the new chromosomal *M. hapla* assembly are due to differences between the genome structure of VW9 and other parental strains. Therefore, the current assembly likely corresponds to a largely accurate representation of the chromosomal DNA molecules in strain VW9.

**Characterization of predicted secreted protein gene repertoire of *M. hapla*.** To annotate the newly generated VW9 genome, we utilized Iso-Seq data from mixed developmental stages, including eggs, J2, and females. This yielded 4,117,943 reads, which were clustered into 240,273 high-quality isoforms with an N50 of 2,018 bp. Additionally, RNA-seq

data were obtained from nematode-infected roots of *Solanum lycopersicum cv* Moneymaker and *S. pimpinellifolium* cv. G1.1554 at five post-inoculation time points (5, 7, 10, 12, and 14 days after inoculation). The integration of these two datasets using BRAKER3 produced a high-quality structural annotation comprising 11,229 protein-coding genes. This gene count is lower than the 14,700 genes reported in the 2008 assembly [24], which was likely due to the reliance in the previous study on ab-initio predictions, which can overestimate gene numbers. The new annotation as assessed with BUSCO using Eukaryota_odb10 [39] showed 88.2% completeness for coding sequences with only 2.4% fragmentation (S4 Table). Similarly, the annotation assessed using BUSCO using Nematoda_odb10 database showed 63.8% completeness for coding sequences with only 2.7% fragmentation. This represents a notable improvement over the 2008 assembly, which showed 78.5% completeness and 10.2% fragmentation with eukaryota_odb10, and 50.5% completeness with 4.1% fragmentation with nematoda_odb10 databases.

To identify genes likely involved in parasitism (candidate effectors), we screened the 11,229 predicted proteins for the presence of secretion signal sequences (SignalP6.0; [40] and the absence of transmembrane domains (DeepTMHMM; [41]. This resulted in the identification of 1,258 genes encoding predicted secreted proteins (PSPs). We then used the full set of 1,258 genes to perform ortholog searches across 71 nematode species using OrthoFinder. Of the 71 nematode species, 33 were plant parasitic nematodes and the rest were either free living or animal parasitic nematodes. This analysis revealed that 1,172 of these PSPs had orthologs in at least one other species while 86 were unique to *M. hapla;* 675 *M. hapla* PSPs were conserved across all 71 nematode species; 401 were present only in RKNs; 3 conserved only with other PPNs; and 1 was shared with only one other nematode species (Fig 6A and 6B and S7 Table). Among the *M. hapla*-specific PSPs, 56 were single-copy genes (singletons) and 30 present in multiple copies.

Of the 1,258 genes encoding PSPs in *M. hapla*, 540 contained known functional domains in Interpro and EggNOG databases, while the remaining 718 did not and are referred to as "pioneer PSPs" (S5 and S6 Tables). For the 1,172 PSPs that had orthologs in other nematodes, 536 contained known domains and 636 were pioneers (Fig 6C and S7 Table). Most of the PSP genes present only in *M. hapla* were pioneers. Of the single-copy PSPs unique to *M. hapla*, 53 had no known functional domains. Three encoded proteins with identifiable features, including a glycine-rich domain, a collagen triple helix repeat, and an SH3 domain. Similarly, among the 30 PSP genes unique to *M. hapla*, but present as multiple copies, only one had a known domain (a protein kinase).

Within the 540 PSP families containing a known functional domain, we identified multiple gene families previously shown to play a role in parasitism in other nematodes (S8 Table). These include several families of CAZymes–31 Glycoside Hydrolases (GH), 17 Glycosyl Transferases (GT), 7 Carbohydrate Esterases (CE), and 12 Pectate Lyases (PL) [42] (S7 Fig and S9 Table). We further verified the annotation of these CAZymes through the dbCAN database [43] (S10 Table). Previous studies suggest that horizontal gene transfers contributed to the acquisition of CAZymes in plant-parasitic nematodes [42]. Using AvP (Alienness vs Predictor) pipeline analysis, we found that 20 CAZymes including 7 GHs, 1 CE and all PLs have evidence of horizontal gene transfer (HGT) (S10 Table). Similarly, other PSPs (Alginate Lyases, Lytic transglycosylases, Fungal chitosanase, Protein Kinase domain containing protein, and Trypsin) also showed evidence of horizontal transfer (S10 Table). Furthermore, consistent with previous phylogenetic studies that suggest pectate lyases expansion through gene duplication during *Meloidogyne* evolution [21,24], our analyses revealed patterns of clade separation and subsequent expan-sion across all four CAZymes families (S8–S10 Figs).

In our structural annotation pipeline, default filtering criteria were employed, which typically exclude very short open reading frames. In addition, the RNA-seq and Iso-seq datasets used in the annotation process were not optimal for identifying short transcripts, as the sequencing read lengths were often insufficient to detect and reconstruct these smaller genes with confidence. As a result, proteins shorter than 66 amino acids were likely not annotated, potentially omitting small, secreted proteins that may function as plant peptide mimics. To more comprehensively identify such candidates, we reanalyzed the 2008 genome annotation and identified 19 small PSPs. Among these, five had functional annotations, including a C2H2 zinc finger domain protein, a fructosyltransferase, a phosphotransferase system component, a

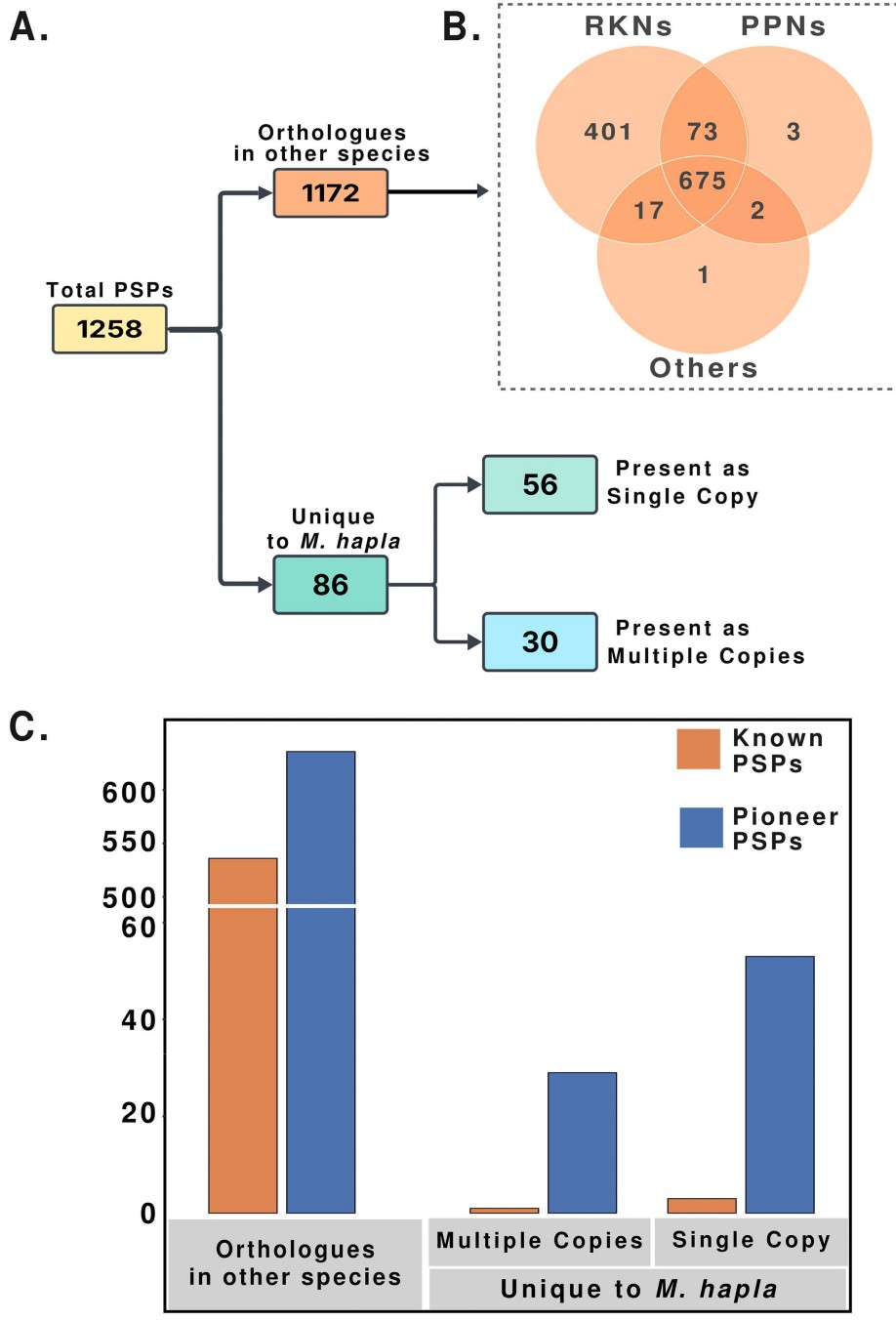

**Fig 6. Classification of *M. hapla* PSPs.** A. Total PSPs were characterized by the presence or absence of orthologs in a set of 71 nematode species. The identification of orthologs was done using Orthofinder. B. Venn diagram shows the PSPs that are shared across RKNs, PPNs and other nematode species. C. The distribution of PSPs into Known and Pioneer PSPs is shown for PSPs with orthologs in other nematode species and for those unique to M. hapla. (i.e., those unique to *M. hapla* having single or multiple copies.

bifunctional nuclease-like protein, and a tyrosinase copper-binding protein. The remaining 14 proteins lacked known functional annotations (S11 Table). When mapped onto the updated *M. hapla* genome assembly, these PSPs were found to be distributed across multiple genomic regions (S11 Table).

Next, we conducted a BLAST search for known plant peptide mimics, including C-terminally Encoded Peptides (CEPs), Inflorescence Deficient in Abscission (IDA) and Rapid Alkalinization Factors (RALF)—all previously described to modulate parasitism related responses in plants [44–46]. We found that all 12 CEPs were located on S13, while IDA1 and IDA2 were located on S7 and S9, respectively. The three RALFs were located on S1, S5 and S10 (S12 Table). No PSY peptide genes were detected, consistent with prior findings that these genes are restricted to Clade I *Meloidogyne* species [47].

### Genes encoding predicted secreted proteins (PSPs) are enriched in high recombination zones

To examine the genomic distribution of protein-coding genes, we compared their empirical cumulative distribution to a theoretical uniform distribution. Overall, genes appear to be evenly distributed along the chromosomes (S11 Fig). However, certain scaffolds (SF1, SF4, SF6, SF7, and SF13) show noticeable,—though often modest—deviations from this uniform pattern. In these regions, uneven gene distribution is sometimes, but not always, associated with unusual recombination patterns. For instance, S1 separates into two genetic linkage groups in the VW9 × LM cross, while S4 and S12 appear to be physically linked. In contrast, S7 contains only a single region with a markedly higher recombination rate. This pattern of uniform gene distribution is consistent with observations in other holocentric species, such as spider mite (*Tetranychus urticae*) and *C. elegans*, where genes are generally evenly distributed along the chromosomes [48,49].

Given the rapid evolution of effectors under selection pressure in various plant pathogens [50,51], we assessed whether PSP genes cluster in HRZs or LRZs (S2 Table). A hypergeometric test for enrichment found that PSP genes were significantly enriched in HRZs ($p = 2.5*10^{-8}$; S13 Table). A scaffold-specific analysis found PSP enrichment on S2, S3, S14, and S16 while other scaffolds did not show significant differences in PSP distribution between HRZs and LRZs (Fig 7 and S14 Table). In addition, annotated PSP families–Cysteine-rich secretory protein, Papain cysteine proteases, Aspartyl proteases, C-type lectin, peptidases, CAZymes, Catalases, Peroxidases, Astacins, Lipases and SXP/RAL2 family–were predominantly localized within HRZs (Fig 8). One notable case is HRZ of S3, which exhibits the highest recombination rate in the genome at 139 cM/Mb (Fig 9A and S2 Table). This HRZ spans 340 KB, and contains 60 genes, 23 (~38%) of which encode PSPs (Figs 9B and 8C). Among these, 16 are pioneer PSPs, while the remaining include known PSPs such as lysozyme, carboxylesterases, glycoside hydrolases, Galectin and Calycin (Lipocalin) (Fig 9C and 9D).

In *C. elegans*, evolutionarily conserved genes are enriched in low recombination regions of chromosomes [52]. To assess whether *M. hapla* shows a similar pattern, we identified 5610 *C. elegans* orthologs from the Orthofinder analysis. Of these, 4195 map to the LRZ and 1415 to the HRZ. An enrichment analysis revealed no overrepresentation of *C. elegans* orthologs in HRZs; however, these orthologs were significantly enriched LRZs ($p = 1.9*10^{-5}$; S13 Table).

### Discussion

Like other *Meloidogyne* species, *M. hapla* lacks telomerase and canonical telomere-associated proteins [27], suggesting that it relies on an alternative mechanism to maintain chromosome ends. In this study, we generated a chromosome-scale genome assembly for the *M. hapla* strain VW9. This assembly consists of 16 scaffolds, corresponding to the previously reported chromosome number for this strain. One notable finding of this study is the identification of tandem repeats consisting of a consensus 16-mer sequence (CCCAAGGTTTAAAAGG) at one or both ends of most scaffolds, and intriguingly, also in an interior region of Scaffold 1. These repeats show no recognizable sequence similarity to the more complex terminal repeats recently identified in Clade I *Meloidogyne* species [26,27] or to the terminal repeats in the diploid species *M. graminicola* [26]. This lack of homology suggests that chromosome end repeat motifs have significantly diversified among RKN species. Notably, most of the 16-mer repeats in our assembly occur at one end of the scaffold, a finding further supported by FISH analysis, which detects these repeats at a single chromosome end. At the ends of the scaffolds

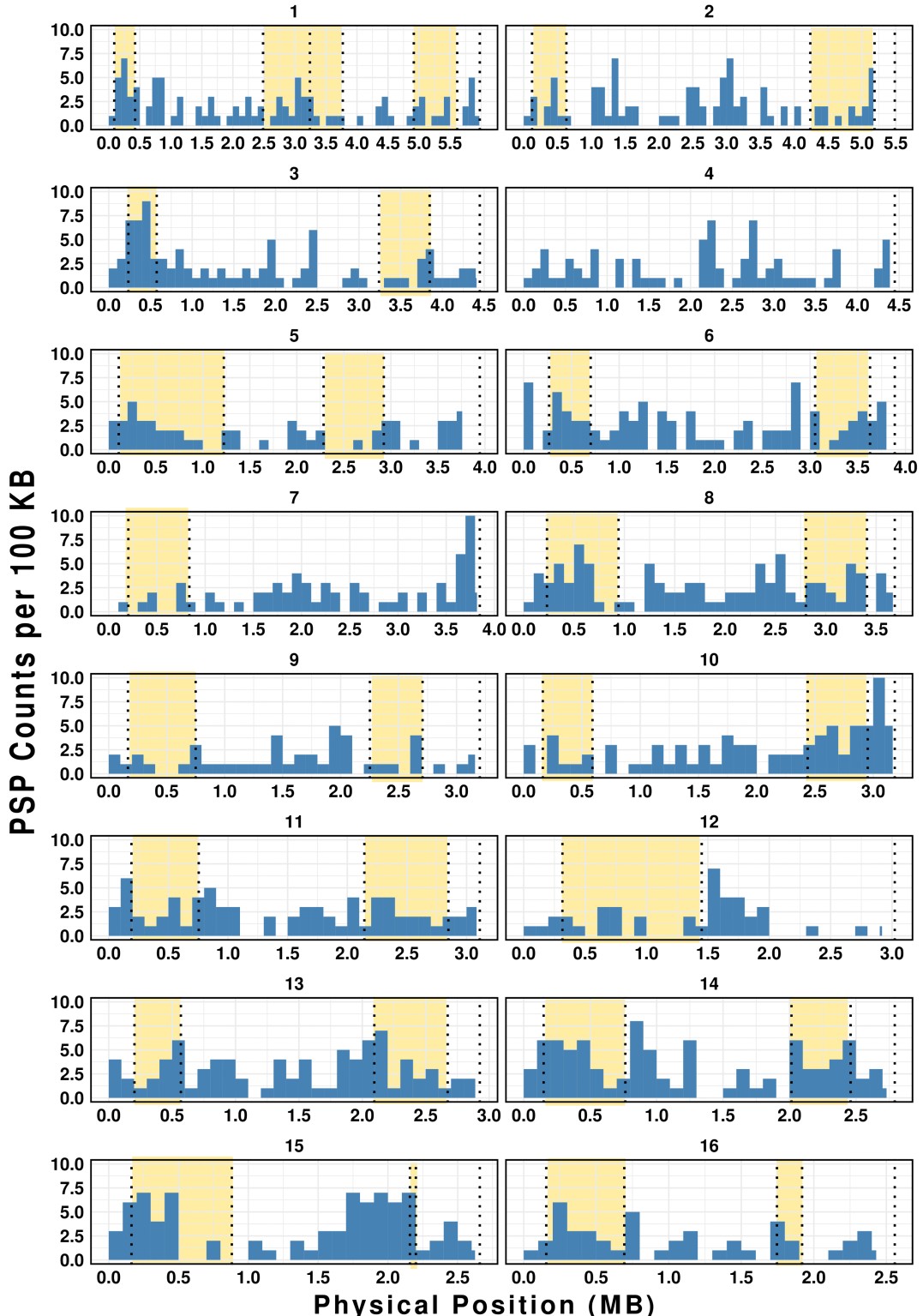

**Fig 7. Distribution of PSP genes in the High Recombination Zones (HRZs).** Each of the 16 scaffolds is divided into 100 kb bins (X-axis). Y-axis represents the number of PSPs per bin. HRZ regions are highlighted in yellow. Genome-wide enrichment analysis shows significant enrichment of PSPs in the HRZs (Hypergeometric test, *p-value = 2.5\*10^{-8}*).

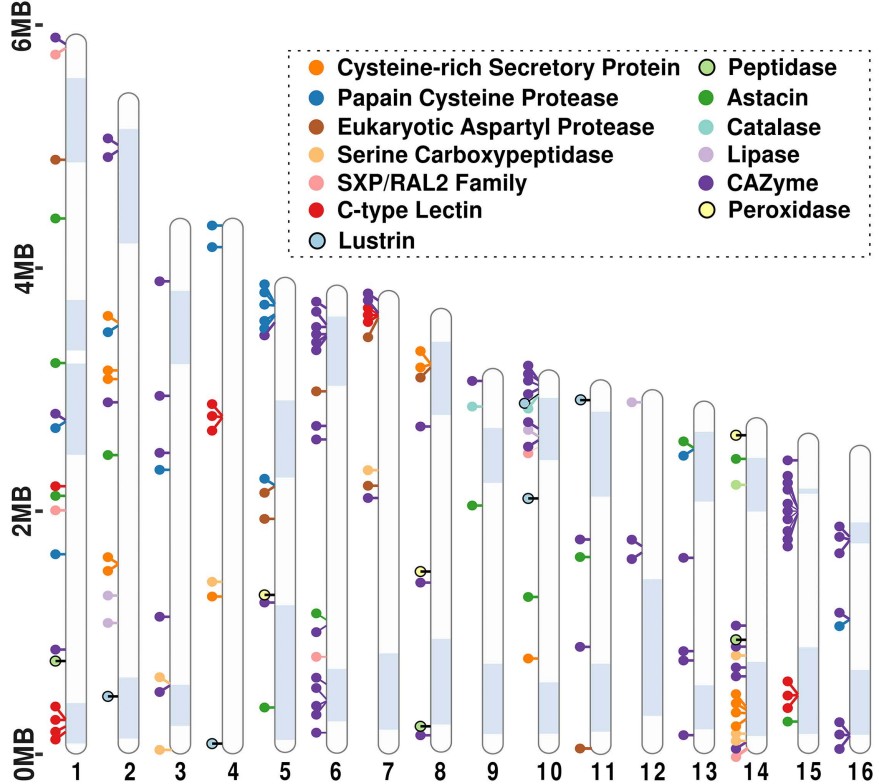

**Fig 8. Distribution of PSPs with conserved domains along the genome of _M. hapla_: Ideogram of PSPs with known functional domains along the scaffolds of _M. hapla_ genome.** Each circle represents the PSPs, and they are pinned with respect to their positions on the scaffolds. The highlighted blue regions denote the HRZs.

where the consensus 16-mer repeat was not detected, we identified two distinct repeat sequences. However, their chromosomal locations have not yet been verified by FISH analysis. Additionally, we cannot rule out the possibility that our assembly is incomplete and that some terminal sequences remain unresolved. The presence of repeats only at one end of the chromosome is reminiscent of _C. elegans_ pairing centers, which facilitate homologous pairing and crossover during meiosis [53]. Whether the terminal regions of _M. hapla_ chromosomes play a role in telomere maintenance, meiotic pairing, or both remains to be investigated.

_M. hapla_ is capable of facultative sexual reproduction and therefore, we can perform controlled crosses and track allele segregation in RIL-like F2 lines. These data provide a valuable resource for studying chromosome structure and parasitism-related adaptations in plant-parasitic nematodes. The recombination patterns of individual F2 lines along the majority of scaffolds in _M. hapla_ resemble those observed in _C. elegans_, with crossovers primarily localized to the arms. However, the overall recombination rate in _M. hapla_ is roughly five times higher than that in _C. elegans_, whose meiotic crossovers are restricted to one per chromosome per meiosis [37,54,55]. The _M. hapla_ genetic map suggests that the one crossover per-chromosome limit may also apply to this species. Therefore, its higher recombination rate could be primarily due to its higher chromosome number and smaller genome size. Recombination in _M. hapla_ is localized to more narrowly defined regions with sharp boundaries, resulting in very high local recombination rates. Intriguingly, electron microscopy studies have described the presence of synaptonemal complexes and distinctive electron-dense "recombination nodules" in oocytes of a meiotic race of _M. hapla_ [56].

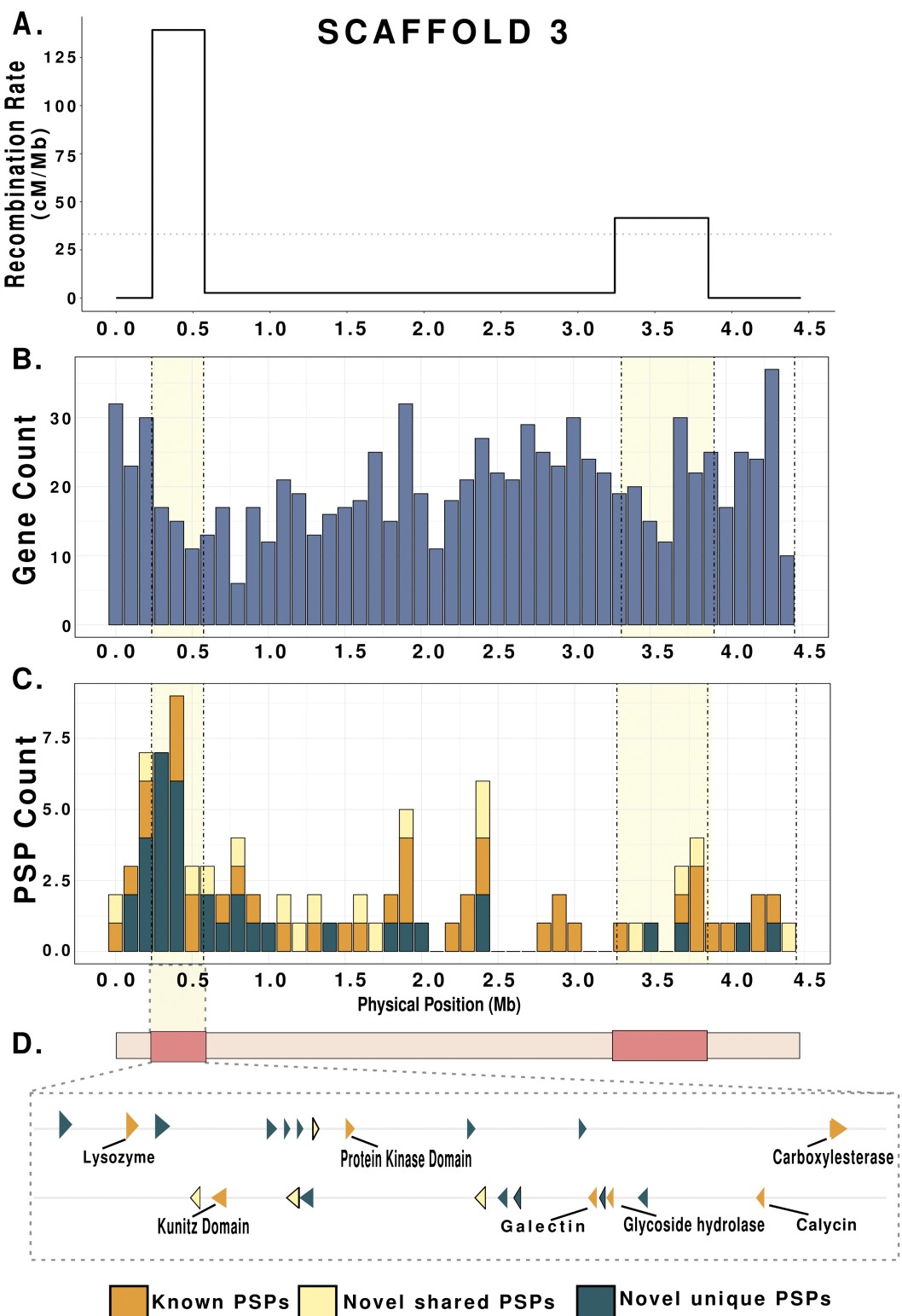

**Fig 9. High recombination zones and gene distribution in Scaffold 3.** A. Recombination rate (cM/Mb) of Scaffold 3 along its physical position (Mb) showing sharply defined HRZs. The dotted horizontal line indicates the average recombination rate for the whole chromosome. B. Histogram of Gene Count per 100 Kb along Scaffold 3. The highlighted regions denote HRZs. C. Stacked bar graph of PSP Count per 100 Kb along the physical positions of scaffold 3 where the highlighted regions denote HRZs. Each color represents functionally annotated PSPs, pioneer PSPs shared with other *Meloidogyne*

species and pioneer PSPs that are unique to *M. hapla*. D. Highlights of the PSP genes found in HRZ of Scaffold 3. They are colored according to the ones shown in C. The arrows represent positive and negative strands. The known PSPs are labelled with their respective functional annotations.

Comparing the genome assembly scaffolds for *M. hapla* strain VW9 and the genetic maps derived from $F_2$ crosses (VW9 × LM and VW8 × VW9) revealed differences in genome rearrangements among the three nematode strains (**Table 1**). The parental strains were derived from geographically distinct field isolates: VW9 from California (U.S.A), VW8 from the Netherlands, and LM from France [6,23]. The reduced recombination observed in S4, S7, and S12 in the VW9xLM cross (**Figs 4** and **5**) is typical of crosses between parents with genomic differences, such as translocations and inversions. S1 exhibited particularly intriguing segregation patterns. Both our assembly and ultra-long sequencing reads support the notion that S1 is a single chromosome. However, F2 progeny from the VW9 × LM cross displayed high recombination rates in the central region of this chromosome for progeny from two of three F1 females (females EA and GC), whereas progeny from the third female (female FB) exhibited no recombination (**Fig 4**). This finding suggests that S1 may exist as a single chromosome in female FB but as two separate chromosome-like entities in females EA and GC. Notably, the classical linkage map from the VW8 × VW9 cross indicates that scaffold S1 is linked to another scaffold (S9), forming a single chromosome. Such variability aligns with earlier global surveys that identified differences in chromosome number and modes of reproduction among *M. hapla* field isolates [17]. Indeed, a cross between females with 15 chromosomes and males with 17 chromosomes produced viable progeny possessing more than 15 chromosomes, suggesting that the chromosome number is fluid within this species [18]. Holocentric chromosomes, such as those found in Nematoda, might be particularly tolerant of fragmentation and fusion events compared to monocentric chromosomes. Additionally, a non-canonical mechanism for chromosome end maintenance might further facilitate this genomic plasticity.

Some, but not all, segregation anomalies that we observed in F2 lines are consistent with inversions, translocations, or other differences in chromosome structure between the parental lines. Intriguingly, for F2 lines of the VW9xLM cross, alleles on Scaffold 13 were strongly skewed toward strain LM (**Fig 4**), but no such bias was observed in VW8xVW9 F2 lines. This strongly biased segregation pattern matches the patterns for toxin-antidote elements observed in multiple species, including the nematode species *C. elegans* and *P. pacificus* [57–60]. The occurrence of such elements hampers overall reproductive success and strongly favors the presence of the toxin-antidote allele in the offspring. In this case, the data are consistent with the presence of a toxin-antidote element in strain LM. Further analysis and additional genetic crosses should provide insight into other observed segregation anomalies.

In our annotation analysis, we focused on PSP genes, as they likely include most of the genes involved in parasitism. Most PSP genes in *M. hapla* had orthologs in other *Meloidogyne* species (**Fig 6**). Additionally, most PSP genes that encode proteins with known domains including all genes with signatures of acquisition via HGT are shared across RKN species. This pattern suggests that the HGT events that introduced these genes occurred before the speciation of these nematodes. Furthermore, over half of the PSP genes that *M. hapla* shares with other RKN species have no identified functional motifs, and nearly all PSP genes that are unique to *M. hapla* are pioneers. Perhaps these unique pioneer PSP genes originated via *de novo* gene birth or represent orphan genes derived from unknown mechanisms [61].

The availability of genetic segregation data provides a unique opportunity to investigate the relationship between recombination rates and the genomic distribution of parasitism genes in meiotic RKN species. In some plant pathogenic fungi, effector genes are often enriched in chromosomal regions associated with high recombination rates. For example, recombination hotspots harbor effector genes in the fungal pathogen *Blumeria graminis* [62]. Similarly, in *Fusarium graminearum* and *Zymoseptoria tritici*, effector genes are enriched in chromosome regions near subtelomeres or on accessory chromosomes; these regions are characterized by higher recombination rates and are described as having a "two-speed" genome architecture [63,64]. Our genome-wide analysis of PSP gene distribution found that these genes are enriched in HRZs (**Fig 7**). By contrast, conserved orthologs shared with C. elegans were enriched in LRZs. Thus, enrichment in HRZs appears to be specific to PSP genes rather than a general feature of gene localization in *M. hapla*.

Double-strand DNA breaks produced during meiosis have been shown to promote ge-nomic variation in humans [65]. In line with this finding, we suggest that the en-richment of PSP genes in HRZs may facilitate diversification, enabling this pest to evade host recognition or expand host range. Interestingly, Clade I *Meloidogyne* species, which reproduce asexually and lack meiosis, and therefore meiotic recombination, also exhibit broad host ranges and can evolve to avoid host resistance. These ameiotic species are of hybrid origin and carry two divergent subgenomes [26,29,66], sug-gesting that their ability to acquire new specificity may be due to interactions between the subge-nomes [66]. Thus, the diploid, meiotic *M. hapla* and the asexual Clade I spe-cies likely achieve adaptability through fundamentally different genomic mecha-nisms. Under-standing how recombination and genome architecture drive PSP evolution could provide new insights into host–parasite co-evolution and support the development of control strategies for nematode infestations. Future studies will focus on comparative genomics and functional analysis to identify the evolutionary processes underlying PSP diversification in these important plant pathogens.

## Materials and methods

### High molecular weight DNA extraction

*Meloidogyne hapla* strain VW9 [6] was propagated on tomato (*Solanum lycopersicum*) cultivar VFNT cherry in a UC Davis Greenhouse facility. Nematode eggs (~1 million) were collected from roots then cleaned by sucrose floatation method and flash frozen in liquid nitrogen. High molecular weight (HMW) gDNA extraction was carried out at UC Davis Genome Center. Two ml of lysis buffer containing 100 mM NaCl, 10 mM Tris-HCl pH 8.0, 25 mM EDTA, 0.5% (w/v) SDS and 100 µg/ml Proteinase K was added to the tube containing frozen eggs. The samples were mixed with gentle pipetting and homogenized at room temperature overnight. The lysate was then treated with 20 µg/ml RNAse at 37 °C for 30 minutes. The lysate was cleaned with equal volumes of phenol/chloroform using phase lock gels (Quantabio Cat # 2302830). The DNA was precipitated from cleaned lysate by adding 0.4X volume of 5M ammonium acetate and 3X volume of ice-cold ethanol. The DNA pellet was washed with 70% ethanol twice and resuspended in an elution buffer (10mM Tris, pH 8.0). The purity of gDNA was assessed using NanoDrop spectrophotometer and 260/280 ratio of 2.0 and 260/230 of 2.29 were observed. For PacBio HiFi, DNA yield was 8 µg as quantified by Qubit 2.0 Fluorometer (Thermo Fisher Scientific, MA). The integrity of the HMW gDNA was verified on a Femto pulse system (Agilent Technologies, CA) where 68% of the DNA was found in fragments above 100 Kb.

### PacBio HiFi sequencing for genome assembly

The HiFi SMRTbell library was constructed using the SMRTbell prep kit 3.0 (Pacific Biosciences, Menlo Park, CA, Cat. #102-182-700) according to the manufacturer's instructions. HMW gDNA was sheared to a target DNA size distribution between 15–20 kb using Diagenode's Megaruptor 3 system (Diagenode, Belgium; cat. B06010003). The sheared gDNA was concentrated using 1X of SMRTbell cleanup beads provided in the SMRTbell prep kit 3.0 for the repair and a-tailing incubation at 37°C for 30 minutes and 65°C for 5 minutes, followed by ligation of overhang adapters at 20°C for 30 minutes, cleanup using 1X SMRTbell cleanup beads, and nuclease treatment at 37°C for 15 minutes. The SMRTbell library was size selected using 3.1X of 35% v/v diluted AMPure PB beads (Pacific Biosciences, Menlo Park, CA; Cat. #100-265-900) to progressively remove SMRTbell templates <5kb. The 15 – 20 kb average HiFi SMRTbell library was sequenced at UC Davis DNA Technologies Core (Davis, CA) using one 8M SMRT cell (Pacific Biosciences, Menlo Park, CA; Cat #101-389-001), Sequel II sequencing chemistry 2.0, and 30-hour movies on a PacBio Sequel II sequencer. This way, we generated 232,679 sequences with an average length of 14,391 base pairs.

### Nanopore sequencing for genome assembly

For Nanopore sequencing, the sequencing libraries were prepared from 1.5µg of high molecular weight DNA using a ligation sequencing kit SQK-LSK114 (Oxford Nanopore Technologies, Oxford, UK). The manufacturer's library preparation

protocol was followed apart from extended incubation times for DNA damage repair, end repair, ligation and bead elution. For sequencing, the PromethION device P24 was used. Thirty fmol of the final library from the sample was first loaded on the PromethION flow cell R10.4.1 (Oxford Nanopore Technologies, Oxford, UK) and run was set up using PromethION MinKNOW 22.12.5 for 17 hours. Data was base called live during sequencing with super-high accuracy mode using ONT-guppy-for-promethion 6.4.6. This way, we generated 584,625 reads with an average length of 27,317 base pairs.

### RNA extraction for Iso-Seq sequencing

Total RNA was extracted from three life stages of *M. hapla*: eggs, newly hatched juveniles, and female nematodes dissected from tomato roots 21 days post infection. RNAeasy mini kit was used as per the manufacturer's instruction for RNA extraction (Qiagen, catalog nr. 74106). To eliminate genomic DNA from RNA samples, TURBO DNAase (Life Technologies, AM1907) treatment was administered. The concentration and purity of the extracted RNA were determined using Nanodrop One Microvolume UV-Vis Spectrophotometer, and RNA integrity and quality were evaluated using bioanalyzer.

### Iso-Seq sequencing for genome annotation

cDNA and SMRTbell library were constructed using SMRTbell prep kit 3.0 (PN 102-396-000 REV02 APR2022) according to manufacturer's instructions. The cDNA was synthesized using NEBNext Single Cell/Low Input cDNA Synthesis & Amplification Module (New England Biolabs, Ipswich, MA, Cat. #E6421L). After fifteen cycles of PCR for cDNA amplification, the cDNA was purified using 0.86X SMRTbell cleanup beads. Subsequently, the cDNA libraries derived from each sample were pooled and used for SMRTbell library construction using the SMRTbell prep kit 3.0 (Pacific Biosciences, Menlo Park, CA, Cat. #102-182-700) with the following enzymatic steps: repair and a-tailing, ligation of overhang adapters, and nuclease treatment. The Iso-Seq SMRTbell library was sequenced at UC Davis DNA Technologies Core (Davis, CA) using one 8M SMRT cell (Pacific Biosciences, Menlo Park, CA; Cat #101-389-001), Sequel II sequencing chemistry 2.0, and 24-hour movies on a PacBio Sequel II sequencer.

### RNA extraction for short-read time-series RNA-seq

Total RNA from the early stage of nematode infection was extracted from galls of tomato-infected roots at five different time points post inoculation. Fourteen days after sowing, tomato plants of *Solanum lycopersicum* cv. Money Maker and *Solanum pimpinellifolium* were inoculated in-vitro with either 0 or 200 surface-sterilized J2s of *M. hapla* strain VW9 (as described in [67]. At 5, 7-, 10-, 12-, and 14-days post inoculation (dpi), all galls present in a single square dish-plate were dissected and flash frozen in liquid nitrogen for RNA-seq. In non-infected plants, to reduce the variation in development, roots segments adjacent to the roots of infected plants were dissected as control. Per time points, two to three technical replicates were taken per plant per nematode treatments.

Liquid nitrogen flash-frozen root galls were smashed and homogenized using Tissuelyzer (Qiagen, Hilden). RNA extraction was performed using the Maxwell 16 LEV-plant RNA kit (Promega) following the manufacturer's instruction. After isolation, RNA concentration and purification was evaluated using NanoPhotometer and spectrophotometer (IMPLEN, CA, United States). RNA integrity checking, library preparations, RNA sequencing and quality filtering was done using BGISEQ-500 at BGI TECH SOLUTIONS (Hongkong) with at least 50 million clean paired-end reads of 150 bp per sample.

### HiC sequencing for scaffolding

Chromatin conformation capture data was generated using a Phase Genomics (Seattle, WA) Proximo Hi-C 4.5 kit, which is a commercially available version of Hi-C Protocol [68]. For library preparation, approximately 500 mg tissue was finely chopped and then crosslinked for 15 minutes at room temperature with end-over-end mixing in 1 mL of Proximo

crosslinking solution. Crosslinking reaction was terminated with a quenching solution for 20 minutes at room temperature again with end-over-end mixing. Quenched tissue was rinsed once with 1X Chromatin Rinse Buffer (CRB). The tissue was transferred to a liquid nitrogen-cooled mortar and ground to a fine powder. Powder was resuspended in 700 µL Proximo Lysis Buffer 1 and incubated for 20 minutes with end-over-end mixing. A low-speed spin was used to clear the large debris and the chromatin containing supernatant transferred to a new tube. Following a second higher speed spin, the supernatant was removed and the pellet containing the nuclear fraction of the lysate was washed with 1X CRB. After removing 1X CRB wash, the pellet was resuspended in 100 µL Proximo Lysis Buffer 2 and incubated at 65°C for 15 minutes.

Chromatin was bound to Recovery Beads for 10 minutes at room temperature, placed on a magnetic stand, and washed with 200 µL of 1X CRB. Chromatin bound on beads was resuspended in 150 µL of Proximo fragmentation buffer and 2.5 µL of Proximo fragmentation enzyme added and incubated for 1 hour at 37°C. Reaction was cooled to 12°C and incubated with 2.5 µL of finishing enzyme for 30 minutes. Following the addition of 6 µL of Stop Solution, the beads were washed with 1X CRB and resuspended in 100 µL of Proximo Ligation Buffer supplemented with 5 µL of Proximity ligation enzyme. The reaction was incubated at room temperature for 4 hours with end-over-end mixing. To this volume, 5 µL of Reverse Crosslinks enzyme was added and the reaction incubated at 65°C for 1 hour. After reversing crosslinks, the free DNA was purified with Recovery Beads and Hi-C junctions were bound to streptavidin beads and washed to remove unbound DNA. Washed beads were used to prepare paired-end deep sequencing libraries using the Proximo Library preparation reagents. HiC sequencing was performed on Illumina (Novaseq) which generated over 52 million read pairs.

### Illumina DNAseq for polishing

DNA extraction from eggs was performed using the CTAB method as described in [26]. KAPA HyperPrep Kit was used according to the manufacturer's protocol for library preparation. This involved processes such as enzymatic fragmentation, end repair, A-tailing, and Illumina-compatible adapter ligation followed by PCR amplification and cleanup using magnetic beads. The final libraries were quantified using Qubit and assessed for size distribution using an Agilent Bioanalyzer. Consequently, the libraries were sequenced on the Illumina NovaSeq X platform using paired-end 150 bp reads. The library preparation and sequencing were performed by Novogene, USA. This way we generated 27 million read pairs.

### Genome profiling of PacBio HiFi reads

The genome profiling of the raw reads generated with PacBio HiFi was conducted using Jellyfish v.2.3.0, Genomescope2 and Smudgeplot0.2.5 [69,70]. K-mer frequencies for the raw reads were determined using Jellyfish with parameters set at "-m 21 -s 1000000000". To deduce the lower and upper coverage range for these reads, "smudgeplot.py cutoff" was utilized. Subsequently, Smudgeplot was generated by using these lower and upper coverage values. Genomescope analysis was performed using default settings with the histogram data generated by Jellyfish as an input.

### Generation of draft assemblies

Multiple draft assemblies of the *M. hapla* genome were made using a combination of sequencing reads using various assembly software. For the final draft assembly that was used to make the final chromosome-scale assembly, we used Hifiasm v.2.0.0 [71]. First, we used Canu to correct the ONT reads to ensure the higher accuracy of ONT reads [72]. After this, we incorporated PacBio HiFi, Canu corrected ONT reads, and raw Hi-C reads into Hifiasm to make a haplotype resolved assembly. The primary assembly generated using this method was polished using Illumina sequencing data in five cycles with Pilon v.1.23 using a Snakemake pipeline (See GitHub) [73]. This corrected 222 bases, introduced 154 insertions, and removed 255 bases. Further, we used Meryl v.1.0 and Merqury v.1.3 to evaluate the quality of the genome assembly and Blobtools2 to check for any contamination in the genome [74–76]. For Blobtools, we followed the protocol outlined in this Github repository in which, we used reference proteomes (release:2024_04) from the UNIPROT database

for diamond BLASTx, and BLAST nt database (2.14), in addition to minimap2 v.2.26-r115 and Samtools v.1.17 [77–80]. For BUSCO analysis, we used eukaryota_odb10 and nematoda_odb10 to check the quality of genome, CDS and protein sequences.

### Chromosome-scale assembly

We used HiC sequencing of VW9 strain to scaffold the draft genome into a chromosome-scale assembly of the *M. hapla* genome. HiC data were generated by Phase Genomics using their Proximo HiC Kit Protocol with four cutter restriction enzymes: DpnII, DDel, Hinfl and Msel. Accordingly, we obtained the restriction sites and ligation junctions of these enzymes and generated restriction enzyme cut sites using the *generate_site_positions.py* script within the Juicer v.1.6. The information of the restriction sites was input within the Juicer v.1.6, and it was run with the early exit parameter [81]. The HiC reads were aligned with the draft genome assembly with bwa alignment v.0.7.17-r1188 [79]. This was followed by quality control and filtering where low quality and chimeric reads were filtered out. The filtered aligned reads were merged into merged_sort.bam file. After the removal of duplicates from the merged file, merged_nodups.txt file was obtained which consisted of interaction matrices. The information obtained from Juicer was used to produce scaffolded assembly using 3D-DNA v.180114 [82]. 3D-DNA uses the interaction matrices to group the contigs into scaffolds. The initial scaffolded output was used to produce a draft contact map. This contact map was manually curated in Juicebox v.1.11.08 [81]. The manual curation involved minor correction of contig positions according to the contact map. The manually curated assembly was then reviewed and finalized using run-asm-pipeline-post-review.sh script from 3D-DNA pipeline and final HiC contact map was made with the sizes of scaffolds in descending order.

### Nigon elements

The output of BUSCO run 'full_table.tsv' was used as an input to visualize the Nigon elements using vis_ALG (https://github.com/pgonzale60/vis_ALG) [33].

### Structural and functional annotation

Before structurally annotating the genome, the draft chromosome-scale assembly was masked using RepeatModeler v.2.0.5 and RepeatMasker v4.1.5 [83,84]. For Iso-Seq pre-processing, we used isoseq3 v.4.0.0 which used ppbam v.2.4.99, pbcopper v.2.3.99, pbmm2 v.1.11.9, minimap2 v.2.15, parasail v.2.1.3, boost v.1.77, htslib v.1.17 and zlib v.1.2.13 [85]. First, we clustered the demultiplexed Full-Length-Non-Chimeric (FLNC) reads using Iso-Seq cluster tool. The high-quality clustered and polished reads were then mapped to the genome using pbmm2 align with preset Iso-Seq parameter. After this, the reads were collapsed using isoseq3 collapse with parameters *–do-not-collapse-extra-5exons, –max-5p-diff5' and –max-3p-diff5'*. These parameters prevent collapsing of isoforms that have extra 5' exons and allow up to 5 base pairs difference at both 5' and 3' ends of the sequences ensuring that transcript diversity was preserved.

For structural genome annotation, we used BRAKER3 installed from its long-read branch [86]. In the first run, BRAKER3 incorporated short-read RNA-seq reads to guide the annotation. In the second run, BRAKER3's long-read protocol was applied during which GeneMarkS-T predicted the protein-coding regions from the long-read transcripts. Finally, TSEBRA was used to combine the data from both short-read and long-read evidence to produce a finalized structural annotation. For functional annotation of the predicted protein-coding genes, we used BLAST2GO from OmicsBox v.3.3.2, InterproScan v.5.72-103.0 and EggNOG v.2.1.12 with eggnog database v.5.02 and novel family database v.1.0.1 [87,88].

### Genetic map based on eSNPs of *M. hapla*

We produced a scaffold-based recombination profile using expressed sequence SNPs (eSNPs) previously identified between 98 RIL-like F lines of *M. hapla* produced by crossing strains VW9 and LM [20]. The transcriptome data in that

study was produced using RNA extracted from *Medicago truncatula* root galls infected with each RIL-like line and was made available from GEO under accession numbers PRJNA229407 and SRP078507. We used hisat2 (v. 2.2.1) to align the reads to the newly assembled chromosome-level genome. Thereafter reads were sorted, duplicate reads were removed, and the reads were indexed using Samtools (v. 1.14). Quality and alignment statistics were generated using fastqc (v. 0.12.1), Samtools, and mosdepth (v. 0.3.3). We used multiqc (v. 1.14) to inspect the qualities per sample. Next, variants were called using bcftools (v. 1.16) with standard settings except (ploidy = 2, keep_alts = true, min_mq = 20, min_bq = 13, min_idp = 5, max_idp = 500, min_qual = 20, min_ad = 2). This generated a file with 58,114 variants, which were further quality filtered using custom scripts in R (v. 4.2.2).

Next, we filtered the variants based on occurrence over the genotypes. We expressed variant occurrence as alternative allele frequencies (fraction of alt calls over total calls). When filtering for at least one clear alternative allele call we found 15,248 sites with at least one alt call. We then removed sites with more than 30% of samples uncalled, leaving 8,455 variants that were evenly distributed over the chromosomes. We called crossovers based on change-point analysis using the change-point package (v. 2.2.4) in R. Per RIL per chromosome, we first interpolated the uncalled sites based on the median genotype of the adjacent five variants before and after the uncalled variant. Subsequently, we used cpt.mean with the method BinSegto identify crossovers. The crossovers were visually inspected for accuracy.

Guo et al. [20] reported that some of the RIL-like lines appeared to be heterozygous. We confirmed this and therefore removed data from EA18, EA30, FB15, FB22, GC4, GC5, GC6, GC31, and GC45 before further analysis. Furthermore, for five lines the coverage by RNA-seq was very low, requiring too much imputation to be of use (EA16, FB19, 12_GC14, GC36, and GC46). This left us with genetic data for 84 RIL-like lines. The set of markers for these lines was pruned for informative markers (markers at either side of a recombination event). The genotype at the ends of the chromosomes were interpolated from the first and last called marker per RIL. In total, this yielded a set of 789 markers for the 84 RILs (S15 Table).

For recombination analysis we first calculated the genetic distances in centimorgans using a custom R script. We analysed physical versus the genetic distance using change-point package in R, we then identified transition points between the domains for each linkage group by performing a change-point analysis using the binary segmentation method [89].

## Integration of genetic linkage maps with chromosome-scale assemblies

To produce a framework genetic linkage map for chromosome-scale assembly, we used segregation data from 458 SNPs in 93 RIL-like F2 lines produced from previously described cross between *M. hapla* strains VW9 and LM [20]. Previously, this segregation data was used to produce a *de novo* genetic map using the previous genome assembly of *M. hapla* obtaining 19 linkage groups [21,24]. To identify the SNP positions in our genome assembly, we extracted sequences spanning 250 bp both upstream and downstream of each of the 458 SNPs using Bedtools v.2.31 [90]. These sequences were subjected to a BLAST against chromosome-scale assembly. We then manually located the positions of the SNPs in the chromosome-scale assembly which was used to align chromosome-length scaffolds to the genetic linkage groups. The same strategy was used to anchor 182 genetic markers developed from an earlier genetic cross in which *M. hapla* strain VW8 as female parent was crossed with VW9 to the current genome assembly [21].

## Terminal repeat regions identification

We ran Tandem Repeats Finder (TRF v4.09) on the terminal 24 kb of each chromosome and recovered a single 16-bp tandem repeat at the 3′ ends of 17 scaffolds [30]. Notably, this was the only significant repeat detected in terminal regions. We then inspected scaffold S1 and identified a similar, but more degenerate, copy of the same repeat near its centre. Thereafter, we scanned the entire set of scaffolds to search for the occurrence of this conserved repeat. However, only S1 contained the 16mer repeat at its center.

To identify additional end-associated repeats, we extracted the first and last 2kb of the 16 scaffolds and used these as input to MEME (classic mode, any-number-of-repetitions), employing a first-order background model to account for the low GC content [31]. We requested up to five frequently occurring motifs with lengths of 30–60 nt. After the MEME run, we analysed motif locations and frequencies with MAST (default parameters) and selected the subset of sequences that contained the most frequent motif. We then performed a second MEME run on this subset, allowing retrieval of a single motif and adjusting the motif length according to the first run to refine the model. Sequences corresponding to occurrences of this refined motif were aligned with MAFFT (–auto), an HMM profile was built with hmmbuild (HMMER), and the profile was searched against the whole genome with nhmmer to verify specific enrichment of repeat arrays at scaffold ends. We iterated this workflow on the remaining sequences (32 – n) until no further runs were necessary and all retained motifs were verified to form repeat arrays at scaffold termini.

Finally, we validated chromosome-end repeats directly in raw Oxford Nanopore reads using TeloSearchLR [32]. We analysed base-called FASTQ files with default parameters except for the k-mer limits (10–40 nt) and terminal window sizes, which we set to 1 kb and 5 kb in separate runs. This analysis confirmed the 16-mer as the top-ranked terminal motif; other motifs detected were either variants of the 16-mer or non-terminal repeats not enriched at chromosome ends. The TeloSearchLR results were used to verify the terminal motifs identified by TRF and MEME.

### Predicted secreted proteins (PSPs) filtering

We used SignalP v.6.0h on slow sequential and DeepTMHMM v.1.0.42 to predict the signal peptides and transmembrane domains in the protein-coding genes respectively [40,41]. We used custom R scripts employed in R v.4.4.1 and RStudio build 421 and python scripts employed in python v.3.10.8 to filter out the genes with Signal Peptide and without any transmembrane domains. The custom R and python scripts can be obtained in this GitHub repository.

### Ortholog analysis

Orthofinder v. 2.5.4 was used to find orthologs of all annotated proteins in *M. hapla* against 72 nematode species with Tardigrade as outgroup [91]. We used the option -msa to produce multiple sequence alignments and phylogenetic trees for all OrthoGroups. We started from a previous OrthoFinder study [27] and we replaced the former *M. hapla* set of predicted proteins by this new one.

### Functional annotation of CAZymes

For functional annotation of CAZymes (GH, PL and GT), we used dbCAN3 using DIAMOND, HMMER via CAZY and dbCAN-sub [92] and we further verified the annotations with InterproScan and EggNOG mapper. Carbohydrate Esterases (CEs) were not annotated by dbCAN3, so we used the predictions from BLAST2GO, InterproScan and EggNOGmapper to annotate these proteins.

### Horizontal gene transfer (HGT) analysis

For HGT analysis, we used the AvP (Alienness vs Predictor) pipeline with a database constructed from UNIREF90 to select, extract and detect the HGT candidates among the predicted secreted proteins [93].

### Phylogenetic trees for CAZymes

For construction of phylogenetic trees, we first used Clustal Omega for the multiple sequence alignment of each CAZyme family with their respective outgroups. Then we used Iqtree to construct the Maximum Likelihood tree under the best-fit model with ultrafast bootstrap that generated 1000 pseudo-replicates to assess support for every internal node. The command used was: iqtree -s input_alignment.phylip -B 1000. Finally, the tree was uploaded to iTOL to make further customizations.

## Hypergeometric test

This analysis was done using R (v4.2.2) using tidyverse (v1.3.2) and purr (0.3.4) packages (Github link). The chromosomal domain content ("HRZ" and "LRZ") was based on the genetic linkage map (S2 Table). Gene start and end positions, obtained from genome annotation, were matched to chromosomal domains by ensuring that a gene's entire length falls within a single domain interval. Genes overlapping multiple domains were assigned to each matching domain, and genes not directly overlapping any domain were assigned to the nearest domain based on the minimum distance between gene boundaries and domain boundaries. We then calculated the total number of genes in each domain (N_domain) and total number genes (N_total). After this, we extracted PSPs per domain (k_domain) and total number of PSPs genome-wide (k_total). We followed this with an enrichment test, where we tested enrichment of PSPs in each domain using a one-tailed hypergeometric test (R's *phyper* function), with the null hypothesis that PSPs are randomly distributed across domains. For a give domain (LRZ/HRZ), the test conducted using significance threshold of α = 0.05

$$P(X \geq k\_domain) = phyper(k\_domain - 1, N\_domain, N\_total - N\_domain, k\_total, lower.tail = FALSE)$$

where, N_domain = number of genes in domain
  k_domain = number of PSPs in domain
  N_total = total number of genes
  k_total = total number of PSPs. Same process was repeated for *C. elegans* orthologs.

## Mitochondrial genome assembly and annotation

In the draft assembly, Contig 35l, with a coverage of 1750x was identified as mitochondrial DNA. After manually removing the tandem duplications at the contig ends and rotating it, we annotated this genome using MITOS2 from web based Galaxy v.24.14 [94]. However, we were unable to fully annotate the mitochondrial genome as the program could not detect multiple core and tRNA genes. Hence, to assemble and annotate a linear mitochondrial DNA, we used MitoHiFi [95] and MITOS2. Again, multiple tRNAs were not identified. Finally, we used MitoZ v.3.6 with Illumina DNA sequences [96]. We used Mitoz all commands with clade option Nematoda and genetic code 5 followed by annotation using the command Mitoz annotate. Although this process gave us the core genes in the mitochondrial genome, it also failed to annotate most of the tRNA genes. We then used MITOS2 to reannotate the linear mitochondrial genome [94]. MITOS2 was able to annotate 10 out of 22 tRNAs. tRNAscan-SE [97] also to detect the remaining tRNAs. We then downloaded the mitochondrial tRNA genes from *M. graminicola* from GenBank accession number KJ139963.1 and blasted these sequences against the linear mitochondrial genome of *M. hapla*. This approach identified 17 out of 22 tRNA sequences. Non-canonical tRNA structures (e.g., armless tRNAs) which are common in nematodes [98] and likely contributed to our difficulties in identifying all the expected tRNA sequences. We then employed Circos to produce a diagram of the mitochondrial genome.

## DNA fluorescence in situ hybridization (FISH) of the candidate terminal repeat

*M. hapla* genomic DNA was isolated from 50 collected egg sacs using Qiagen Blood and Tissue kit (Qiagen) according to the manufacturer's procedure with slight modifications. Tissue was homogenized for 30 sec using electric homogenizer and pestle and incubated overnight at 56 °C. Sample was then treated with RNase A for 5 min at RT and final elution was in 100 µL of the elution buffer. For PCR procedures obtained DNA was diluted to concentration of 0,02 ng/µL. To produce probes corresponding to the 16 bp repeat (GTTTAAAAGGCCCAAG) identified at scaffold ends, we extended primers to encompass more than one monomer to avoid primer dimers as much as possible (Mhap_tel_4R AAGATTTAAAAGGC-CCAAGATTTAAAAGAC Mhap_tel_4L CTTTTAAACCTTGGGTCTTTTAAACCTTG). FISH probes were prepared according to the previously described procedure [99] with 61.7 °C as annealing temperature. Prepared probes were cleaned

using QIAquick PCR Purification Kit (Qiagen) and eluted in 50 μL of nuclease-free water. They were tested on 1% agarose gel to check their concentration and length (S12 Fig).

Whole *M. hapla* females were squashed onto slides and different fixatives were tested to optimize chromosome morphology. Ethanol and acetic acid fixative proved to be optimal for both preservation of chromosome morphology and FISH analyses and slide preparation was done as described [100]. The FISH procedure was done according to established protocols [27,99] except for skipping the pretreatment step of specimens in 45% acetic acid as here the slides were fixed in a mixture already containing acetic acid. Similarly, as *M. incognita*, it was not possible to count chromosome numbers exactly and prometaphases/metaphases are rare. Telomeric signals had different intensity from very large to very discrete and consequently it was hard to count the precise number of certain signals. Best evaluation was obtained on elongated chromosomes (prometaphase).

## Supporting information

**S1 Fig. Blobplot/Contamination plot of draft assembly.**
(TIF)

**S2 Fig. Mitochondrial genome of Meloidogyne hapla strain VW9.**
(TIF)

**S3 Fig. Genome Profile of Meloidogyne hapla strain VW9.**
(TIF)

**S4 Fig. HiC quality check and HiC assembly with ONT reads.**
(TIF)

**S5 Fig. Three groups of chromosome-end repeats found in M. hapla genome.**
(TIF)

**S6 Fig. Genetic correlation between and within scaffolds.**
(TIF)

**S7 Fig. Distribution of CAZymes in the genome of M. hapla.**
(TIF)

**S8 Fig. Phylogenetic tree and distribution of GH proteins.**
(TIF)

**S9 Fig. Phylogenetic trees and distribution of CE and PL proteins.**
(TIF)

**S10 Fig. Phylogenetic tree and distribution of GT proteins.**
(TIF)

**S11 Fig. Distribution of genes across the genome of M. hapla.**
(TIF)

**S12 Fig. Agarose gel image of prepared FISH probe with λ DNA for concentration assessment of markers.**
(TIF)

**S1 Table. Chromosome-end repeats, their positions and copy numbers in each scaffold.**
(XLSX)

**S2 Table. Recombination Data per Scaffold.**
(XLSX)

**S3 Table. Marker positions from two crosses: VW8 X VW9 and VW9 X LM in VW9 scaffolds.**
(XLSX)

**S4 Table. BUSCO scores for genome, coding sequence (CDS) and protein coding genes.**
(XLSX)

**S5 Table. Functional annotation of Predicted Secreted Peptides (PSPs) using EggNOG and InterProScan.**
(XLSX)

**S6 Table. Functionally un-annotated PSPs and their positions in M. hapla genome.**
(XLSX)

**S7 Table. List of orthologs and unique genes in M. hapla (Orthofinder Results).**
(XLSX)

**S8 Table. Classification of major known PSPs according to their functional annotation.**
(XLSX)

**S9 Table. Ideogram data used for CAZymes.**
(XLSX)

**S10 Table. Annotation of CAZymes from dbCAN3 server and their HGT evidence according to AvP tool.**
(XLSX)

**S11 Table. Small PSPs from [24] assembly and their positions in the new genome.**
(XLSX)

**S12 Table. Known plant peptide mimics and their distribution in M. hapla genome.**
(XLSX)

**S13 Table. Hypergeometric test for enrichment of PSPs and C. elegans orthologs in high and low recombination zones of M. hapla scaffolds.**
(XLSX)

**S14 Table. Hypergeometric test for enrichment of PSPs in each scaffold of M. hapla genome.**
(XLSX)

**S15 Table. Genotypes of RILs and corresponding markers.**
(XLSX)

## Acknowledgments

We would like to thank Michael Winters and Dave Lunt for helpful discussions. We would also like to thank the University of California Davis farm server system and Wageningen University HPC cluster system for their support with the data analysis.

## Author contributions

**Conceptualization:** Pallavi Shakya, Etienne G. J. Danchin, Evelin Despot-Slade, Nevenka Meštrović, Valerie M. Williamson, Mark G. Sterken, Shahid Siddique.

**Data curation:** Pallavi Shakya, Mark G. Sterken, Shahid Siddique.

**Formal analysis:** Pallavi Shakya, Etienne G. J. Danchin, M. Laurens Voogt, Stefan J. S. van de Ruitenbeek, Adam P. Taranto, Ana Zotta Mota, Mark G. Sterken.

**Funding acquisition:** Mark G. Sterken, Shahid Siddique.

**Investigation:** Pallavi Shakya, Muhammad I. Maulana, Jacinta Gimeno, Alison C. Blundell, Evelin Despot-Slade, Nevenka Meštrović.

**Methodology:** Pallavi Shakya, Muhammad I. Maulana, Etienne G. J. Danchin, Jacinta Gimeno, Adam P. Taranto, Alison C. Blundell, Evelin Despot-Slade, Ana Zotta Mota, Dadong Dai, Valerie M. Williamson, Mark G. Sterken, Shahid Siddique.

**Project administration:** Mark G. Sterken, Shahid Siddique.

**Resources:** Valerie M. Williamson, Shahid Siddique.

**Supervision:** Valerie M. Williamson, Shahid Siddique.

**Validation:** Pallavi Shakya, Stefan J. S. van de Ruitenbeek.

**Visualization:** Pallavi Shakya, Evelin Despot-Slade, Nevenka Meštrović.

**Writing – original draft:** Pallavi Shakya, Mark G. Sterken, Shahid Siddique.

**Writing – review & editing:** Pallavi Shakya, Muhammad I. Maulana, Etienne G. J. Danchin, M. Laurens Voogt, Stefan J. S. van de Ruitenbeek, Jacinta Gimeno, Adam P. Taranto, Alison C. Blundell, Evelin Despot-Slade, Nevenka Meštrović, Ana Zotta Mota, Dadong Dai, Valerie M. Williamson, Mark G. Sterken, Shahid Siddique.

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
