## [Decision Letter · Decision Letter 0]

30 Sep 2025

PPATHOGENS-D-25-01755

High-Resolution Genome Assembly and Linkage Mapping in Meloidogyne hapla Reveal Non-Canonical Telomere Repeats and Recombination Hotspots Associated with Effector Proteins

PLOS Pathogens

Dear Dr. Siddique,

Thank you for submitting your manuscript to PLOS Pathogens. After careful consideration, we feel that it has merit but does not fully meet PLOS Pathogens's publication criteria as it currently stands. Therefore, we invite you to submit a revised version of the manuscript that addresses the points raised during the review process.

Please submit your revised manuscript within 30 days Nov 29 2025 11:59PM. If you will need more time than this to complete your revisions, please reply to this message or contact the journal office at plospathogens@plos.org. Please include the following items when submitting your revised manuscript:

We look forward to receiving your revised manuscript.

Kind regards,

Adler R. Dillman, Ph.D.

Academic Editor

PLOS Pathogens

Shou-Wei Ding

Section Editor

PLOS Pathogens

Sumita Bhaduri-McIntosh

Editor-in-Chief

PLOS Pathogens

orcid.org/0000-0003-2946-9497

Michael Malim

Editor-in-Chief

PLOS Pathogens

orcid.org/0000-0002-7699-2064

**Additional Editor Comments :**

This is an excellent paper which makes a significant contribution to the field. After review by three experts in the field, only minor revisions are recommended; this is a compliment to care the authors took in writing the first draft. There was one major concern raised about tandem repeat elements, but the remainder of the concerns were considered minor. I believe the authors will be able to address all of the issues raised in a timely manner. I look forward to seeing a revised version of this manuscript.

**Journal Requirements:**

**Reviewers' Comments:**

Reviewer's Responses to Questions

**Part I - Summary**

Reviewer #1: Shakya et al. took advantage of multiple sequencing strategies to build the first chromosomal-level genome assembly of the a plant parasitic nematode (PPN), RKN M. hapla. This is a significant update to the highly fragmented draft genome published in 2008, and will be of valuable resource to the PPN community. I am satisfied with the critical analyses (FISH, telomeric repeat, genetic linkage) the authors conducted to support the validity of their chromosomal scaffolds. This study also highlights the possibility that meiotic recombination may have resulted in effector diversification necessary for host-parasite interactions; this would certainly be interesting to the broad readership of PLoS Pathogens. Overall, this manuscript is well written, includes novel bioinformatic and biological data, and represents a significant stride forward for the discipline.

Reviewer #2: Shakya et al. describe the assembly and analysis of a chromosomally-scaffolded genome assembly for the plant-parasitic root-knot nematode (RKN) Meloidogyne hapla. Their work provides a substantially improved resource for molecular genetic analysis and possible control of this major RKN and, indirectly, related Meloidogyne species. Their work also reveals noteworthy phenomena of chromosomal organization, telomeric repeat distribution, unequal meiotic recombination rates along chromosomes, and the partial tendency of genes encoding putative effectors of infection to be disproporationately distributed in genomic regions of elevated recombination. The quality of this work leaves me with only one significant scientific question, along with some minor questions and suggested corrections.

Reviewer #3: Shakya et al. have produced the most contiguous genome assemby of Meloidogyne hapla, an important plant-parasitic species. They have used all the latest sequencing technologies (Illumina, PacBio, ONT, and Hi-C) resulting in 16 genomic scaffolds and 1 mitochondrial contig. They have also identified a non-canonical 16-mer telomeric repeat which they validated with FISH experiments. This falls in line with the presence of different non-canonical repeats in other Meloidogyne species, making the genus an interesting case for telomere analyses. They then used F2 lines from a cross of two different M. hapla strains to produce recombination frequence profiles and together with the genome annotation show that predicted secreted proteins (PSPs) are enriched in high recombination zones (HRZs). Finally, they argue that maybe PSPs in HRZs are enabling the parasitism of a broad host range and the evasion of host recognition.

I find the genome assembly to be of the highest quality. Clearly, considerable thought, effort, and rigorous analysis were devoted to this project. Having high-quality genome assemblies for plant parasitic species enables future research into plant-parasitism, comparative genomics, and in this case structural analyses.

The article is well written, and I highly recommend it for publication. However, I have some minor comments for the authors to consider.

**Part II – Major Issues: Key Experiments Required for Acceptance**

Reviewer #1: None

Reviewer #2: There are no experiments that I would *require* in order to validate the study's conclusions. There are experiments that I would suggest to strengthen its conclusions, which I describe in the section on "Minor Issues".

Reviewer #3: (No Response)

**Part III – Minor Issues: Editorial and Data Presentation Modifications**

Reviewer #1: P16, L18 (6th line from the bottom): Change ‘virulence’ to ‘parasitism’

P18, L3: Remove the ‘- (hyphen)’ from ‘Simi-larly’

P19, L11: Remove the repetitive words: ‘show noticeable’

P24, 3rd line from the bottom: Remove the repetitive sentence: ‘This observation is further supported by FISH analysis, which detects these repeats at a single chromosome end.’

P42, L9: Change ‘O/N’ to ‘overnight’

Reviewer #2: TANDEM REPEAT ELEMENTS

My one significant scientific question is about the ascertainment of short tandemly repeated sequence elements. The authors observed a novel 16-mer forming tandem repeats at the ends of several chromosomes (as well as in the center of chromosomal scaffold 1). Given the way the text is written, they apparently observed *no* other short tandemly repeated elements at the ends of chromosomes (e.g., at either end of chromosomal scaffold 1, at the right end of chromosomes 3 and 4, at the left end of chromosome 6, etc.). It is not clear from the Methods whether the Tandem Repeats Finder (TRF) was used genomewide to identify possible tandem repeats, or whether it was run only in the terminal 24 kb of each chromosome. It is also not clear whether the authors did or did not observe any other significant tandemly repeated elements in any chromosomal end regions other than the 16-mer that they describe here (the reader is left to assume that the authors did not). The authors note in the Discussion that "we cannot rule out the possibility that ... additional relevant sequences remain undetected".

I would like the authors to do the following:

1. Explicitly state in their Results whether TRF did, or did not, identify any other tandemly repeated elements other than the 16-mer in the terminal 24 kb of chromosomal scaffolds. If TRF did identify such elements, add them to their existing Supp. Table 1.

I would also like the authors to strongly consider doing the following:

2. In addition to their existing analyses, use an alternative method to identify telomere-associated tandemly repeated elements directly from their long-read sequence data (which they have in abundance -- they have 56x genomic coverage of high-accuracy HiFi reads, as well as 143x coverage of less accurate Oxford Nanopore reads). One recently published method that the authors could use for this analysis is TeloSearchLR.

TeloSearchLR code: https://github.com/gchchung/TeloSearchLR

TeloSearchLR bioconda package: https://bioconda.github.io/recipes/telosearchlr/README.html

TeloSearchLR PubMed ID: https://pubmed.ncbi.nlm.nih.gov/40169380

TeloSearchLR paper: https://academic.oup.com/g3journal/article/15/6/jkaf062/8102967

By running TeloSearchLR, the authors should rediscover the 16-mer as a positive control result. But in addition, any other tandem repeats associated with telomeres (e.g., at either end of chromosomal scaffold 1) may also be discoverable by TeloSearchLR. Having such a result could make their analysis of the Meloidogyne hapla VW9 genome more complete.

MINOR QUESTIONS AND SUGGESTED CORRECTIONS

In the Introduction:

"Meloidogyne spp. exhibits" should read ""Meloidogyne spp. exhibit" ('spp.' abbreviates '[two or more] species', which will take a plural verb).

In the Results:

"...with the consensus sequence (CCCAAGGTTAAAAGG) at 17 scaffold ends." This sentence and the paragraph it starts are not incorrect as they stand, but the way they were organized (and the fact that no mention at all of variant tandem repeats is made in the legend of Figure 1) initially confused me badly when I read this, and if left unchanged may confuse readers as well. To make things clearer, I would suggest revising the first sentence to read ".....with the consensus sequence (CCCAAGGTTAAAAGG) at 17 scaffold ends, and with one variant at an 18th scaffold end." The rest of the paragraph can then be left unchanged but readers should be less easily confused.

"at the center of S1 scaffold" should read "at the center of the S1 scaffold".

Figure 1 shows the distribution of tandem repeats in all chromosomes, and *includes* the repeats at the left end of scaffold S10, but does not tell (or remind) the reader that S10's tandem repeats are of a variant repeat unit. An additional sentence should be added to the legend of Figure 1B to make this obvious. It might say: "Except for scaffold S10, all repeats have a 16-mer consensus element (shown); those in S10 have a variant element."

"Of the 1258 genes encoding PSPs..." should (I think) read "Of the 1,258 genes encoding PSPs..."

"p-value=2.5e-08" with the '08' in superscript could be better written either by not putting the '08' in superscript (which would at least make the text more readable) or by instead writing out the numbers as "p-value = 2.5•10-8".

Reviewer #3: 1) I find the mitochondrial genome assembly workflow to be a bit strange. The authors identified contig 351 as mitochondrial. However, they then assembled the mitochondrion de novo using only the Illumina reads with MitoZ. Why didn't the authors polish contig 351, or alternatively, why didn't they utilise the available long reads? Regarding the tRNA annotation, did the authors compare with the mitochondria of other Meloidogyne species to check how many tRNAs they have?

2) The Nigon analysis is not integrated into the overall flow of the article. I would assume that the genomic structural variability even between M. hapla strains (as shown in the next section) is responsible for the Fig. 3 result. Maybe the authors can add a few sentences to connect these two findings.

3) There are multiple good-quality Meloidogyne assemblies available. Have the authors considered to use them for comparative structural analyses?

4) The Iso-Seq and RNA-seq data were used only for annotation purposes. Have the authors considered examining expression patterns - if any - and mapping them to the PSPs inside and outside of the HMZs?

# Typos (P=Page, L=Line)

P.18 - L. 3 : Simi-larly to Similarly

- L. 5 : Ta-ble to Table

P.19 - L. 11 : "show noticeable" is written twice

P.24 - : "a finding further supported by FISH analysis, which detects these repeats at a single chromosome end" is written twice

PLOS authors have the option to publish the peer review history of their article (what does this mean? ). If published, this will include your full peer review and any attached files.

**Do you want your identity to be public for this peer review?** For information about this choice, including consent withdrawal, please see our Privacy Policy .

Reviewer #1: No

Reviewer #2: No

Reviewer #3: No

**Figure resubmission:**
---

## [Editor Report · Decision Letter 1]

10 Nov 2025

Dear Professor Siddique,

We are pleased to inform you that your manuscript 'High-Resolution Genome Assembly and Linkage Mapping in Meloidogyne hapla Reveal Non-Canonical Telomere Repeats and Recombination Hotspots Associated with Effector Proteins' has been provisionally accepted for publication in PLOS Pathogens.

Best regards,

Adler R. Dillman, Ph.D.

Academic Editor

PLOS Pathogens

Shou-Wei Ding

Section Editor

PLOS Pathogens

Sumita Bhaduri-McIntosh

Editor-in-Chief

PLOS Pathogens

orcid.org/0000-0003-2946-9497

Michael Malim

Editor-in-Chief

PLOS Pathogens

orcid.org/0000-0002-7699-2064

Thank you for your careful revision and thoughtful response to the reviewers' concerns. The revised version is an improvement on what was already a solid investigation of the M. hapla genome.
---

## [Editor Report · Acceptance letter]

Dear Professor Siddique,

We are delighted to inform you that your manuscript, "High-Resolution Genome Assembly and Linkage Mapping in Meloidogyne hapla Reveal Non-Canonical Telomere Repeats and Recombination Hotspots Associated with Effector Proteins," has been formally accepted for publication in PLOS Pathogens.

Best regards,

Sumita Bhaduri-McIntosh

Editor-in-Chief

PLOS Pathogens

orcid.org/0000-0003-2946-9497

Michael Malim

Editor-in-Chief

PLOS Pathogens

orcid.org/0000-0002-7699-2064